# Water-resistant redox-active metal–organic framework

Ryota Akai [1], Showa Kitajima[1], Kohei Okubo [1], Nobuyuki Serizawa [2], Hirotomo Nishihara [1,3], Hitoshi Kasai[1] & Kouki Oka [1,4,5] ✉

Metal–organic frameworks (MOFs) comprise coordination bonds and have attracted attention for electrochemical applications. However, MOFs are usually structurally weak in aqueous solutions, especially in acidic aqueous solutions, owing to their coordination bonds, making their application in charge-storage devices challenging. In the current work, we demonstrate a redox-active MOF (RAMOF) that is structurally stable and achieves reversible charge storage with almost the theoretical capacity even in acidic aqueous electrolytes owing to its strong Zr–O bonds and the large coordination number. In addition, the RAMOF exhibits high durability ( > 98% after 100 cycles) and high Coulombic efficiency (99.9%) owing to its high crystallinity and proton conductivity. An aqueous MOF–air rechargeable battery is fabricated and exhibits high durability and high Coulombic efficiency. Furthermore, the material recycling of the RAMOF based on its coordination bonds is demonstrated. Therefore, we conceptually prove the application and advantages of RAMOFs in aqueous environments.

Metal–organic frameworks (MOFs) are crystalline porous materials, wherein pores of different dimensions and sizes are constructed via coordination bonds between metal ions and organic linkers[1–3]. As the environments and functions of the pores, permeable for electrolytes, can be facilely tuned by changing the components, metals, and organic linkers, MOFs have attracted attention in various applications, such as electrode-active materials[4–7], gas adsorption materials[2,8,9], catalysts[4,10], and sensors[1,11,12]. However, since acidic protons can cleave coordination bonds in MOFs by promoting hydrolysis of the metal-organic linker bonds[13], MOFs are usually structurally weak in aqueous solutions, especially in acidic aqueous solutions, making their application in aqueous charge-storage devices challenging[14,15].

Redox-active MOFs (RAMOFs), which comprise redox-active sites in their structures, exhibit high capacity retention in organic electrolytes owing to their structural stability in organic electrolytes, and therefore have been applied as electrode-active materials for metal-ion batteries[4,5,16]. However, as mentioned above, most RAMOFs undergo structural collapse in aqueous electrolytes, especially in acidic aqueous electrolytes, owing to which their application as electrode-active materials for aqueous batteries is limited to batteries except for acidic aqueous electrolytes[17–22]. In addition, although RAMOFs have a robust three-dimensional structure and pores for effective electrolyte permeation, no RAMOF has been reported to achieve high values for all the three criteria required for application as a charge-storage material in aqueous electrolytes: full capacity (close to the theoretical capacity based on the molecular weight), high durability (capacity retention close to 100%), and high Coulombic efficiency (discharge capacity/ charge capacity close to 100%).

Meanwhile, metal–air rechargeable batteries, which use metals (e.g., aluminum and zinc) as the anode-active material, oxygen as the cathode-active material, and basic aqueous solutions as the electrolyte, have been developed and are expected to achieve high energy densities ascribed to resource-abundant oxygen with high capacity[23–25]. However, dendrites usually precipitate on the surfaces of

[1]Institute of Multidisciplinary Research for Advanced Materials, Tohoku University, Sendai, Miyagi, Japan. [2]Department of Applied Chemistry, Faculty of Science and Technology, Keio University, Yokohama, Kanagawa, Japan. [3]Advanced Institute for Materials Research (WPI-AIMR), Tohoku University, Sendai, Miyagi, Japan. [4]Carbon Recycling Energy Research Center, Ibaraki University, Ibaraki, Japan. [5]Deuterium Science Research Unit, Center for the Promotion of Interdisciplinary Education and Research, Kyoto University, Kyoto, Japan. ✉e-mail: oka@tohoku.ac.jp

the metal anodes during repeated charging and discharging, which degrades the cyclability of the batteries[24]. In addition, highly concentrated basic aqueous electrolytes (6–7 M KOH aqueous solution) are often used for efficient ionic conduction, which causes carbonate clogging owing to reactions of the electrolyte with atmospheric $CO_2$, leading to lower cyclability of the batteries[26]. To solve these problems, organic–air rechargeable batteries that use organic redox-active materials as the anode-active material and acidic aqueous solutions as the electrolyte have been demonstrated[27–32]. These batteries inherently avoided issues such as dendrite precipitation and carbonate clogging that were commonly observed in metal–air rechargeable batteries[27–32]. However, despite these advantages, organic redox-active materials often suffer from gradual dissolution or degradation in acidic aqueous electrolytes during repeated cycling, which still limits their cyclability[33]. In other words, an anode-active material that is able to store charge with high durability (capacity retention close to 100%) even in such electrolytes is highly required.

In the current work, we focus on UiO-66, which has been reported to be a crystalline porous material, a water-resistant MOF, especially an acid-resistant MOF, owing to its strong Zr–O bonds[14,15,34] and the large coordination number[35]. By introducing redox-active $p$-hydroquinone units (redox potential: approximately +0.1 V vs. Ag/AgCl[27,36]), in place of benzene in the organic linker of the acid-resistant MOF, we prepare the acid-resistant RAMOF UiO-66-$(OH)_2$, which achieves reversible charge storage with an ideal capacity close to the theoretical capacity based on the molecular weight even in acidic aqueous electrolytes, owing to its optimized particle size, high porosity, and proton conductivity. In addition, the RAMOF exhibits high durability and high Coulombic efficiency owing to its high crystallinity and proton conductivity. Then, by using the RAMOF as an anode-active material, an aqueous MOF–air rechargeable battery is fabricated. In addition, after use of the battery, we recycle the RAMOF through a simple treatment with an aqueous carbonate solution because the coordination bonds of the RAMOF exhibit instability in aqueous carbonate solutions while retaining robustness in acidic aqueous solutions.

## Results and discussion
### Preparation and characterization of UiO-66-$(OH)_2$
As shown in Fig. 1a, UiO-66-$(OH)_2$ with 1,4-dihydroxybenzene[27,33] as an organic redox-active linker was prepared with reference to a previous work[37]. Since electrode-active materials with low conductivity could only achieve reversible charge storage with an ideal capacity close to theoretical capacity up to 100 nm from the conductive surface[38], as shown in Supplementary Table 1 Entry 1 and 2, UiO-66-$(OH)_2$ with small particle size (average particle size: $70 \pm 20$ nm) was prepared by decreasing the precursors' concentration and reaction time in the microwave. As shown in Supplementary Fig. 1, smaller particle sizes were obtained at lower precursors' concentrations. This trend demonstrated a positive correlation between precursors' concentrations and particle sizes under otherwise identical reaction conditions. In fact, UiO-66-$(OH)_2$ with a large particle size prepared by long reaction time, shown in Supplementary Figs. 2 and 3 and Table 1 Entry 3, did not achieve an ideal capacity close to the theoretical capacity (Supplementary Fig. 4).

In this section, we characterized UiO-66-$(OH)_2$ as a MOF (Supplementary Table 1 Entry 2). As shown in Fig. 1b, c, the Brunauer–Emmett–Teller (BET) surface area and the pore size of UiO-66-$(OH)_2$ were evaluated based on $N_2$ gas adsorption/desorption at 77 K, and was found to be 1075 $m^2 g^{-1}$ and 0.62 nm, respectively, similar to previously reported values[39,40], indicating that the UiO-66-$(OH)_2$ in the current work also had a high porosity and specific surface area. Based on powder X-ray diffraction (PXRD) analysis, as shown in Fig. 1d (blue line), UiO-66-$(OH)_2$ was confirmed to be isostructural with the previously reported UiO-66[41]. As shown in Fig. 1d (red line), the crystallinity of UiO-66-$(OH)_2$ was maintained even after immersion in a

0.05 M $H_2SO_4$ aqueous solution for 24 h; this was presumably because of the Zr–O bonds with high bond energy of $766.1 \pm 10.6$ kJ $mol^{-1}$[14,15,34] and the large coordination number (12 organic linkers per Zr cluster, which was the largest value among MOFs[35]). In order to support that strong Zr–O bonds were formed, we also synthesized UiO-66-$(OH)_2$ under harsher conditions (higher temperature, longer reaction time, and higher precursors' concentration, where the details are provided in the caption of Supplementary Fig. 6) than those previously reported[37], and measured Fourier-transform infrared (FT-IR) spectra (Supplementary Fig. 6). As shown in Supplementary Fig. 6, the peak positions derived from O–Zr–O and Zr–(OC) in UiO-66-$(OH)_2$ prepared under different reaction conditions were identical (662 $cm^{-1}$[42] and 575 $cm^{-1}$[39], respectively), indicating that strong Zr–O bonds were successfully formed, even under the reaction conditions (Experimental Section 2.1), to resist acidic aqueous solutions. As shown in Fig. 1e and Supplementary Fig. 7, the organic linkers of UiO-66-$(OH)_2$ are usually readily missing[39]. Therefore, the percentage of missing organic linkers in UiO-66-$(OH)_2$ was evaluated based on thermogravimetric analysis (the details are described in the Experimental Section 2.3)[43–45]. The compound was found to have 5.91 organic linkers per Zr cluster out of a theoretical organic linker number of 6 per Zr cluster.

### Electrochemical properties of UiO-66-$(OH)_2$
Then, we characterized the electrochemical properties of UiO-66-$(OH)_2$. In the first cycle, the potential was swept from +0.50 V vs. Ag/AgCl to −0.20 V vs. Ag/AgCl and then swept back to +0.50 V vs. Ag/AgCl. As shown in Supplementary Fig. 8, in the potential region, where the redox reaction between $p$-hydroquinone and $p$-benzoquinone was often observed[46–48], the UiO-66-$(OH)_2$/carbon/polyvinylidene difluoride (PVdF) composite electrode exhibited no redox capability. However, as shown in Fig. 2a, upon sweeping the potential in the positive direction from +0.50 V vs. Ag/AgCl to +0.90 V vs. Ag/AgCl, an irreversible oxidation peak appeared at around +0.7 V vs. Ag/AgCl, and a redox peak emerged in the range of −0.1 – +0.4 V vs. Ag/AgCl. In order to investigate the details of the charge storage mechanism of UiO-66-$(OH)_2$, as shown in Supplementary Figs. 9 and 10, we performed ex situ and in situ FT-IR analyses. By focusing on the initial state of the structure of UiO-66-$(OH)_2$, as shown in Supplementary Fig. 11 and Supplementary Table 5, we found that the density functional theory (DFT)-optimized structure of the cluster of UiO-66-$(OH)_2$ exhibited an O···O distance of 2.48 Å[49], suggesting that the initial state of the structure of UiO-66-$(OH)_2$ had hydrogen bonds between the protons of C–(OH) of $p$-hydroquinone and carboxy groups. As shown in Fig. 2a and Supplementary Fig. 10a, an irreversible oxidation peak appeared at around +0.7 V vs. Ag/AgCl upon sweeping the potential in the positive direction from +0.50 V to +0.90 V vs. Ag/AgCl. As shown in Supplementary Fig. 9a, b, in the ex situ FT-IR spectrum after applying the potential at +0.90 V vs. Ag/AgCl, a new peak appeared at 1638 $cm^{-1}$, derived from C=O[50], which indicated that the new peak was attributed to the formation of C=O by the oxidation (Supplementary Fig. 9c). This attribution was further supported by the finding that the oxidation potential shift to the positive direction was presumably caused by hydrogen-bond formation, as reported in previous works[51,52]. In addition, as shown in Supplementary Fig. 10a and b, the peak intensity of 1235 $cm^{-1}$ at the initial state of the red solid-line spectrum, attributed to C–(OH) of $p$-hydroquinone[53–55], decreased upon oxidation, resulting in the purple solid-line spectrum, thereby suggesting the oxidation of hydrogen-bonded C–(OH) of $p$-hydroquinone. From the above results, as shown in Supplementary Fig. 10c, the oxidation peak at around +0.7 V vs. Ag/AgCl was attributed to the oxidation of hydrogen-bonded C–(OH) of $p$-hydroquinone to C=O of $p$-benzoquinone. After that, as shown in Supplementary Fig. 10a, upon sweeping the potential from +0.90 V to −0.20 V vs. Ag/AgCl, we observed a reduction peak at around 0.0 V vs. Ag/AgCl and an increase of the peak intensity from the purple solid- to the blue solid-line spectra (Supplementary Fig. 10b). As

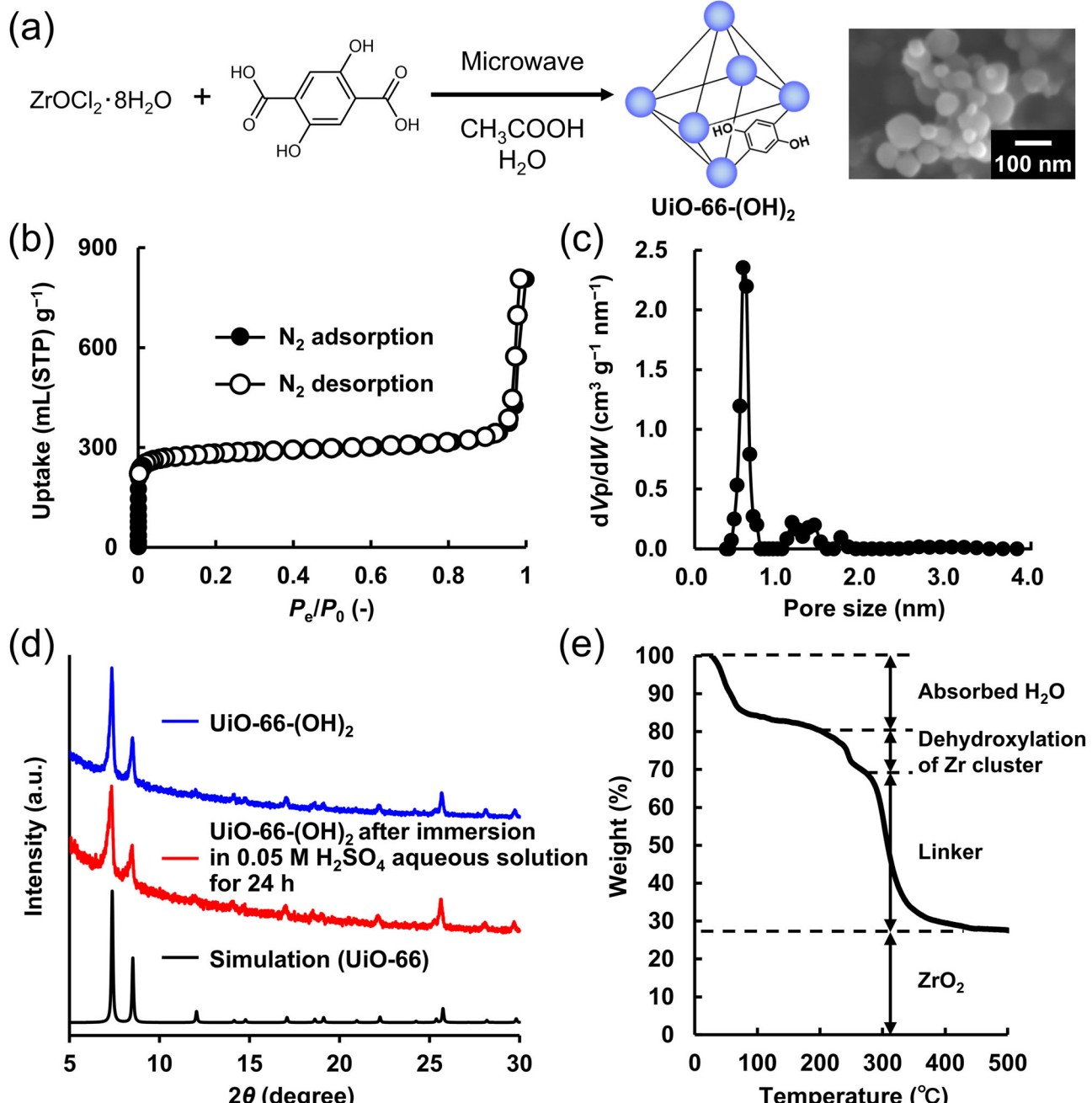

**Fig. 1 | Preparation and characterization of UiO-66-(OH)₂. a** Preparation and scanning electron microscopy (SEM) image of UiO-66-(OH)₂ (scale bar: 100 nm). **b** N₂ adsorption/desorption isotherms of UiO-66-(OH)₂ at 77 K. The crystallinity was maintained after N₂ adsorption (Supplementary Fig. 5). **c** Pore size distribution of UiO-66-(OH)₂ ($V_p$: pore volume, $W$: pore width). **d** Powder X-ray diffraction (PXRD) patterns of UiO-66-(OH)₂ (blue), after immersion in a 0.05 M H₂SO₄ aqueous solution for 24 h (red), and simulation of UiO-66 (black). **e** Thermogravimetric analysis of UiO-66-(OH)₂ under air. UiO-66-(OH)₂ exhibited three-step weight loss: the first step was caused by the loss of absorbed water from the crystal structure, the second step was caused by the dehydration of the Zr clusters, and the third step was caused by the loss of the organic linker (2,5-dihydroxyterephthalic acid)[43-45]. Source data are provided as a Source Data file.

shown in Supplementary Fig. 12 and Supplementary Table 6, the molecular electrostatic potential (MESP) map suggested that C=O of *p*-benzoquinone in the oxidation state of UiO-66-(OH)₂ would be the reduction site for proton storage owing to the strongly negative MESP value of the oxygen atoms in *p*-benzoquinone[56,57]. These results indicated that the reduction peak around 0.0 V vs. Ag/AgCl (Supplementary Fig. 10a) was attributed to the reduction of C=O of *p*-benzoquinone to C−(OH) of *p*-hydroquinone (Supplementary Fig. 10c). After that, upon sweeping the potential from −0.20 V to +0.90 V vs. Ag/AgCl (Supplementary Fig. 10a), as shown in Supplementary Fig. 10b, we

observed two kinds of oxidation peaks at around +0.3 V (polarized) and +0.7 V vs. Ag/AgCl, and observed a decrease in peak intensity from the blue solid- via red dotted- to purple dotted-line spectra. Based on these peak intensity changes (Supplementary Fig. 10b and c) and previous literature (redox reactions of non-hydrogen-bonded *p*-hydroquinone[47,48]), the oxidation peaks around +0.3 V and +0.7 V vs. Ag/AgCl were attributed to non-hydrogen-bonded and hydrogen-bonded C−(OH) of *p*-hydroquinone, respectively. From the above analyses, as shown in Fig. 2a, the oxidation peak at around +0.7 V vs. Ag/AgCl was attributed to the oxidation of hydrogen-bonded C−(OH)

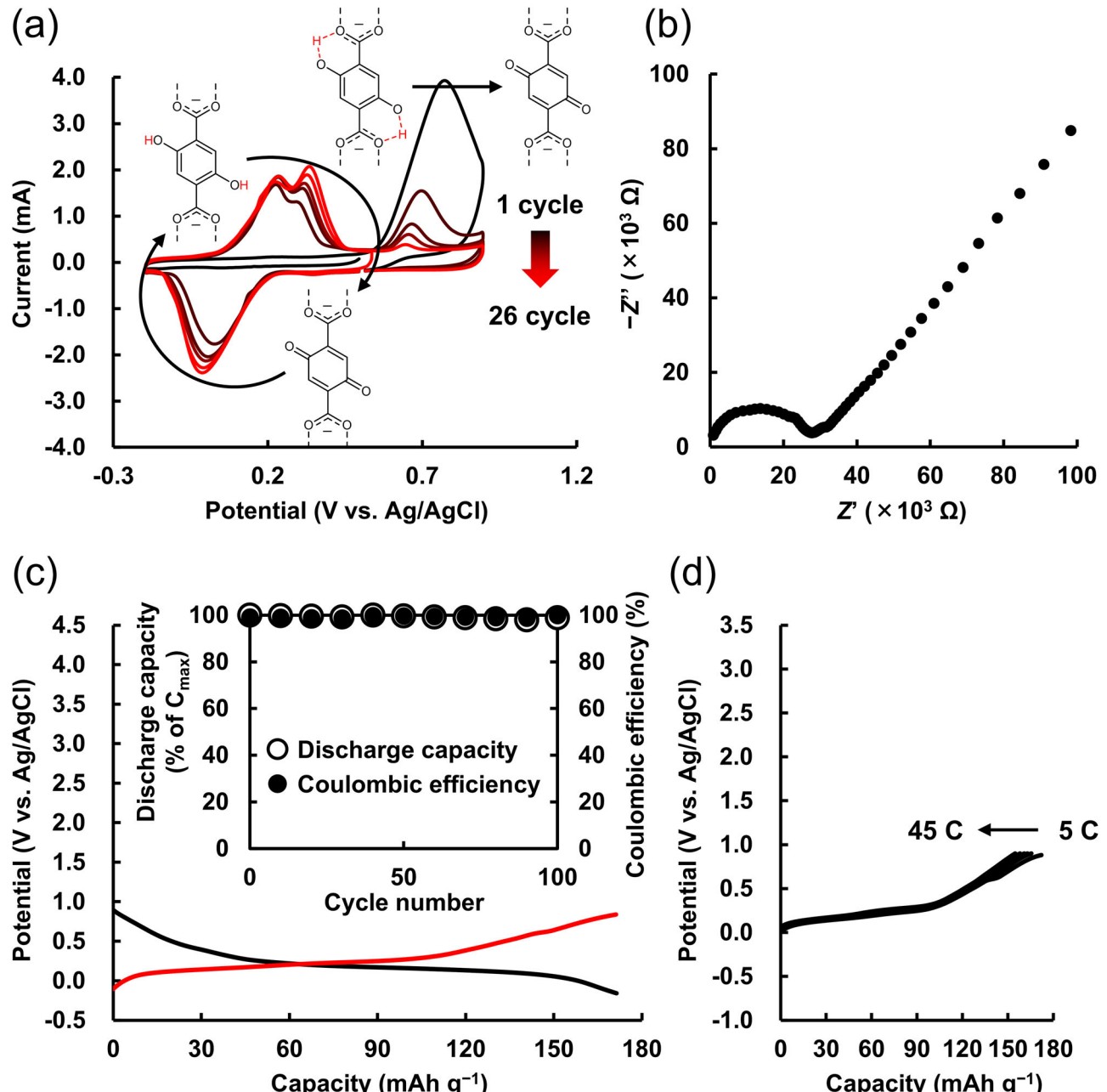

**Fig. 2 | Electrochemical properties of UiO-66-(OH)₂. a** Cyclic voltammogram of the UiO-66-(OH)₂/carbon/PVdF composite electrode in a 0.05 M H₂SO₄ aqueous solution under Ar atmosphere at the scan rate of 10 mV s⁻¹. As shown in Fig. 2a, a polarization was observed. The previous work reported that the semiquinone state is stabilized at pH < 1, according to DFT calculations, leading to two one-electron oxidation steps rather than a single two-electron step[46]. In addition, since the conversion of quinone in its neutral state to quinone radical anion was unfavorable, the reduction proceeded via the protonated intermediate[46]. These factors contributed to the polarization observed in the oxidation process. **b** Cole-Cole plot of UiO-66-(OH)₂. Impedance spectrum of the disk-shaped pellet under 95% RH at 30 °C (Z': real part, Z'': imaginary part). The flattened semicircles represented the bulk and grain boundary resistances. The crystallinity was maintained even after impedance measurements (Supplementary Fig. 16). **c** Charging (black)/discharging (red) curves of half-cell using the UiO-66-(OH)₂/GMS/PVdF composite electrode at 5 C. Inset: The electrode cycle test (42 C). At 42 C, the UiO-66-(OH)₂/GMS/PVdF composite electrode achieved a discharge capacity of more than 90% of the theoretical capacity based on the molecular weight of UiO-66-(OH)₂. Therefore, we performed a cycling test at 42 C. **d** Rate capability of the UiO-66-(OH)₂/GMS/PVdF composite electrode (5, 10, 15, 20, 30, and 45 C). Source data are provided as a Source Data file.

of *p*-hydroquinone (Supplementary Fig. 10c), and the redox peak in the range of −0.1 − +0.4 V vs. Ag/AgCl was attributed to the redox reaction of non-hydrogen-bonded C−(OH) of *p*-hydroquinone (Supplementary Fig. 10c).

As shown in Fig. 2b, the proton conductivity of the pelletized UiO-66-(OH)₂ was measured by conducting electrochemical impedance spectroscopy (EIS), and the proton conductivity was calculated based on a fitting analysis[58] assuming an equivalent circuit (Supplementary

Fig. 13 and Supplementary Table 2). As shown in the flattened semicircles in Fig. 2b, dielectric relaxation was observed in the high frequency range, and the proton conductivity of UiO-66-(OH)₂ was 2.18 × 10⁻⁶ S cm⁻¹ under 95% relative humidity (RH) at 30 °C. As shown in Supplementary Fig. 14, the direct current (DC) electrical conductivity calculated from DC resistance measurement was 3.81 × 10⁻⁸ S cm⁻¹, which was significantly lower than 2.18 × 10⁻⁶ S cm⁻¹, supporting 2.18 × 10⁻⁶ S cm⁻¹ as the proton conductivity of UiO-66-(OH)₂ under

95% RH at 30 °C. The Arrhenius plot in Supplementary Fig. 15 gave an activation energy ($E_a$) of 2.02 eV ( > 0.4 eV[59]), indicating that the proton conduction occurred via the vehicle mechanism. As shown in Fig. 1e, since UiO-66-$(OH)_2$ readily absorbed water molecules[60], protons should be transferred to the interior of the UiO-66-$(OH)_2$ crystals by the vehicle mechanism, owing to which high rate capabilities as an electrode-active material were expected.

Most organic redox materials have low conductivities; therefore, conductive additives are usually required to enable their use as organic electrode-active materials[28,30,61,62]. Accordingly, as shown in Supplementary Figs. 17a and b, two common conductive additives for organic electrode-active materials, Super P Conductive Carbon Black (Super P)[63] and single-walled carbon nanotubes (SWNTs)[64], were tested by fabricating UiO-66-$(OH)_2$/carbon/PVdF composite electrodes (the details are given in the Experimental Section 2.4). Super P had a small particle size ( < 40 nm), which made it difficult to disperse and support UiO-66-$(OH)_2$ on a conducting surface. Meanwhile, the fibrous SWNTs got easily entangled and bundled[65], which also made it difficult to disperse UiO-66-$(OH)_2$, as it was insoluble in organic solvents. As shown in Supplementary Fig. 18 (blue and black lines), the UiO-66-$(OH)_2$/Super P/PVdF composite electrode exhibited a discharge capacity of 17.9 mAh $g^{-1}$ (10% of the theoretical capacity) and the UiO-66-$(OH)_2$/SWNT/PVdF composite electrode exhibited a discharge capacity of 96.1 mAh $g^{-1}$ (56% of the theoretical capacity); that is, both composite electrodes were unable to achieve theoretical capacity based on the molecular weight of UiO-66-$(OH)_2$. Then, we evaluated the redox capability of the UiO-66-$(OH)_2$/graphene mesosponge (GMS)/PVdF composite electrode in a 0.05 M $H_2SO_4$ aqueous solution. GMS is a three-dimensional graphene material with high flexibility, porosity, and conductivity[66,67]. As shown in Supplementary Fig. 17c, UiO-66-$(OH)_2$ was well-dispersed on GMS. As shown in Supplementary Fig. 18 red, the UiO-66-$(OH)_2$/GMS/PVdF composite electrode exhibited a superior redox capability to those of Super P and SWNTs. GMS has rarely been used as a conductive additive for RAMOFs, and was employed in the current work to enhance their electrical conductivity.

As shown in the charge/discharge curves in Fig. 2c, the UiO-66-$(OH)_2$/GMS/PVdF composite electrode exhibited a plateau potential of around +0.15 V vs. Ag/AgCl and a discharge capacity of 171.2 mAh $g^{-1}$, which was close to the theoretical capacity (171.9 mAh $g^{-1}$) estimated from the molecular weight of UiO-66-$(OH)_2$. The electrolyte easily soaked into UiO-66-$(OH)_2$ owing to its high porosity (BET surface area: 1075 $m^2$ $g^{-1}$, pore size: 0.62 nm) and proton conductivity ($2.18{\times}10^{-6}$ S $cm^{-1}$ under 95% RH at 30 °C), and GMS formed a good adhesive interface with UiO-66-$(OH)_2$ owing to its small particle size ( < 100 nm). Therefore, almost all the organic linkers of UiO-66-$(OH)_2$ stored protons and electrons. Furthermore, the Coulombic efficiency achieved 99.9%, which indicated that protons and electrons were reversibly stored owing to the high crystallinity and proton conductivity of UiO-66-$(OH)_2$. As shown in Fig. 2c inset, the UiO-66-$(OH)_2$/GMS/PVdF composite electrode exhibited a high cyclability of more than 98% of its initial capacity even after 100 cycles. The PXRD in Supplementary Fig. 19 confirmed that the structure of UiO-66-$(OH)_2$ was maintained owing to its strong Zr−O bonds and the large coordination number even after 100 cycles in the half-cell, showing its high structural stability. As shown in Supplementary Fig. 20, a long-term cycle test of the electrode was also performed. The UiO-66-$(OH)_2$/GMS/PVdF composite electrode retained over 95% of its initial capacity even after 1000 cycles, demonstrating its high cyclability. In addition, as shown in Fig. 2d, the UiO-66-$(OH)_2$/GMS/PVdF composite electrode exhibited high-rate capabilities based on the proton conductivity ($2.18 \times 10^{-6}$ S $cm^{-1}$ under 95% RH at 30 °C) of UiO-66-$(OH)_2$, achieving a discharge capacity of 154.8 mAh $g^{-1}$ (90% of the theoretical capacity) even at 45 C. Therefore, although MOFs usually decompose in aqueous solutions, particularly in acidic aqueous solutions, we have demonstrated a RAMOF that was structurally stable and achieved reversible charge storage of 171.4 mAh $g^{-1}$ (close to the theoretical capacity based on the molecular weight) even in acidic aqueous electrolytes, while also exhibiting high durability ( > 98% after 100 cycles), and high Coulombic efficiency (99.9%) owing to its high crystallinity and proton conductivity.

## Aqueous MOF−air rechargeable batteries

As shown in Fig. 3a, b, an aqueous MOF−air rechargeable battery with the UiO-66-$(OH)_2$/GMS/PVdF composite electrode as the anode, Pt/C as the cathode, and a 0.05 M $H_2SO_4$ aqueous solution as the electrolyte was fabricated. It should be noted that, until now, RAMOFs have been applied only as cathode-active materials in aqueous rechargeable batteries, whereas the current work demonstrated their use as an anode-active material (Supplementary Table 3). As shown in Fig. 3c, the battery exhibited charging/discharging curves of a Coulombic efficiency of 99.9% and a plateau discharging voltage of around +0.56 V, demonstrating reversible proton and electron storage capability. As shown in Fig. 3c, the discharge capacity of the battery was 171.8 mAh $g^{-1}$, which corresponded to the theoretical capacity (171.9 mAh $g^{-1}$), and therefore almost all organic linker sites reversibly stored protons and electrons (Coulombic efficiency was almost 99.9%). As shown in Fig. 3c inset, the aqueous MOF−air rechargeable battery exhibited a high cyclability of 99% of its initial capacity even after 100 cycles. The PXRD and ex situ FT-IR spectra in Supplementary Figs. 19 and 21 confirmed that the structure and composition of UiO-66-$(OH)_2$ were maintained owing to its strong Zr−O bonds and the large coordination number even after 100 cycles of the battery, supporting that both the structure and composition of UiO-66-$(OH)_2$ remained unchanged. As shown in Supplementary Fig. 22, a long-term battery cycle test was also performed. The battery retained over 92% of its initial capacity even after 1000 cycles, demonstrating its high cyclability. In addition, as shown in Fig. 3d, the battery exhibited high-rate capabilities, achieving a discharge capacity of 157.3 mAh $g^{-1}$ (92% of the theoretical capacity) even at 45 C, and, as shown in Supplementary Fig. 23, it retained a discharge capacity of 102.8 mAh $g^{-1}$ (60% of the theoretical capacity) at 60 C. Figure 4a−d and Supplementary Tables 3 and 4 summarize the advantages of the aqueous MOF−air rechargeable battery compared to aqueous MOF-based rechargeable batteries[17–22] and aqueous organic−air rechargeable batteries[27–32,38,56,62,68–70]. The current work demonstrates a high battery performance; reversible charge storage with an ideal capacity close to theoretical capacity (99.9%), high durability (99% after 100 cycles), and high Coulombic efficiency (99.9%).

## Decomposition and reconstruction of the UiO-66-$(OH)_2$/GMS/ PVdF composite electrode

To demonstrate the advantages of RAMOFs, constructed via coordination bonds, in an aqueous environment, we demonstrated the material recycling of the UiO-66-$(OH)_2$. As the aqueous MOF−air rechargeable battery was simply composed of the UiO-66-$(OH)_2$/GMS/ PVdF composite electrode immersed in an electrolyte, the anode, cathode, and the electrolyte could be easily separated. The UiO-66-$(OH)_2$/GMS/PVdF composite could be facilely stripped from the current collector. As shown in Figs. 5, 1. UiO-66-$(OH)_2$ was decomposed into metals and organic linkers by soaking the composite electrode in a 1 M $NH_4HCO_3$ aqueous solution[71] (Experimental Section 2.2), and GMS and PVdF were separated by filtration to obtain a solution of metals and organic linkers (Supplementary Fig. 24). Following the method in the Experimental Section 2.2, as shown in Supplementary Fig. 25, 2. UiO-66-$(OH)_2$ was reconstructed and recycled (hereinafter referred to as UiO-66-$(OH)_2$-R). As described in the Experimental Section 2.2, although the yield of UiO-66-$(OH)_2$-R (approximately 10%) was still low, the recycling yield could be improved by investigating decomposition and reconstruction conditions (e.g., solvent and modulator) in our continuous work. As shown in Supplementary Fig. 26, UiO-66-$(OH)_2$-R

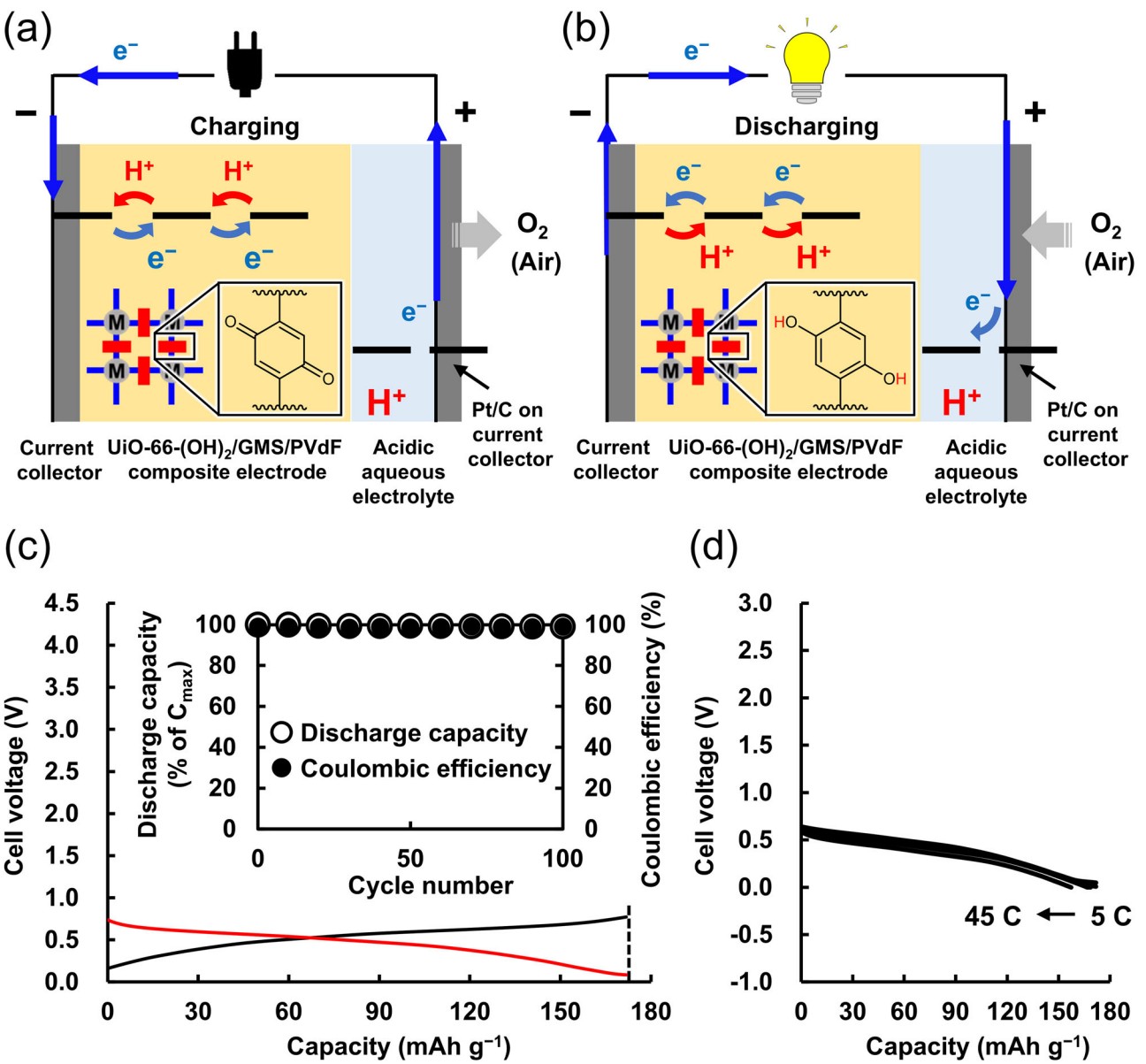

**Fig. 3 | Schematic and performance of the aqueous metal–organic framework (MOF)–air rechargeable battery.** Schematic diagrams of the (**a**) charging/(**b**) discharging of the aqueous MOF–air rechargeable battery. **c** Charging (black)/discharging (red) curves of the aqueous MOF–air rechargeable battery at 5 C. The dotted line represents the theoretical capacity based on the molecular weight of UiO-66-(OH)$_2$ (171.9 mAh g$^{-1}$). Inset: The battery cycle test (14 C). At 14 C, the battery achieved a discharge capacity of more than 98% of the theoretical capacity based on the molecular weight of UiO-66-(OH)$_2$. Therefore, we performed a cycling test at 14 C. **d** Rate capability of the battery (5, 10, 15, 20, 30, and 45 C). Source data are provided as a Source Data file.

became more defective than the original UiO-66-(OH)$_2$, with a theoretical capacity of 154.1 mAh g$^{-1}$. As shown in Supplementary Fig. 27, 3. the UiO-66-(OH)$_2$-R/GMS/PVdF composite electrode exhibited a discharge capacity of 152.9 mAh g$^{-1}$, which corresponded to the theoretical capacity (154.1 mAh g$^{-1}$), successfully proving that UiO-66-(OH)$_2$ could be decomposed and reconstructed (recycled) as an anode-active material. Therefore, as shown in Fig. 5, combined with the decomposition/reconstruction of UiO-66-(OH)$_2$ based on its coordination bonds, the advantages of RAMOFs in an aqueous environment were conceptually demonstrated.

Since MOFs are well-known to be structurally unstable in acidic aqueous solutions, owing to their coordination bonds, the application of RAMOF as electrode-active materials for aqueous batteries has been limited to systems that avoid acidic aqueous electrolytes. In the current work, we demonstrated a high-performance RAMOF with *p*-hydroquinone as an organic linker, whose reversible charge storage

capability was elucidated through ex situ and in situ FT-IR analyses and DFT calculations. The UiO-66-(OH)$_2$ was structurally stable even in acidic aqueous electrolytes owing to its strong Zr–O bonds and the largest coordination number in MOFs, and achieved reversible charge storage with an ideal capacity close to the theoretical capacity of the UiO-66-(OH)$_2$ for the first time by reducing its particle size. In addition, the RAMOF exhibited high durability ( > 98% after 100 cycles) and high Coulombic efficiency (99.9%) owing to its high crystallinity and proton conductivity. An aqueous MOF–air rechargeable battery was fabricated with the RAMOF as the anode-active material; the battery exhibited high durability (99% after 100 cycles) and high Coulombic efficiency (99.9%), which indicated that using a RAMOF as an anode-active material overcame the weak points of aqueous organic-air rechargeable batteries. Furthermore, the material recycling of the RAMOF based on its coordination bonds was demonstrated. Therefore, we conceptually proved the application and advantages of

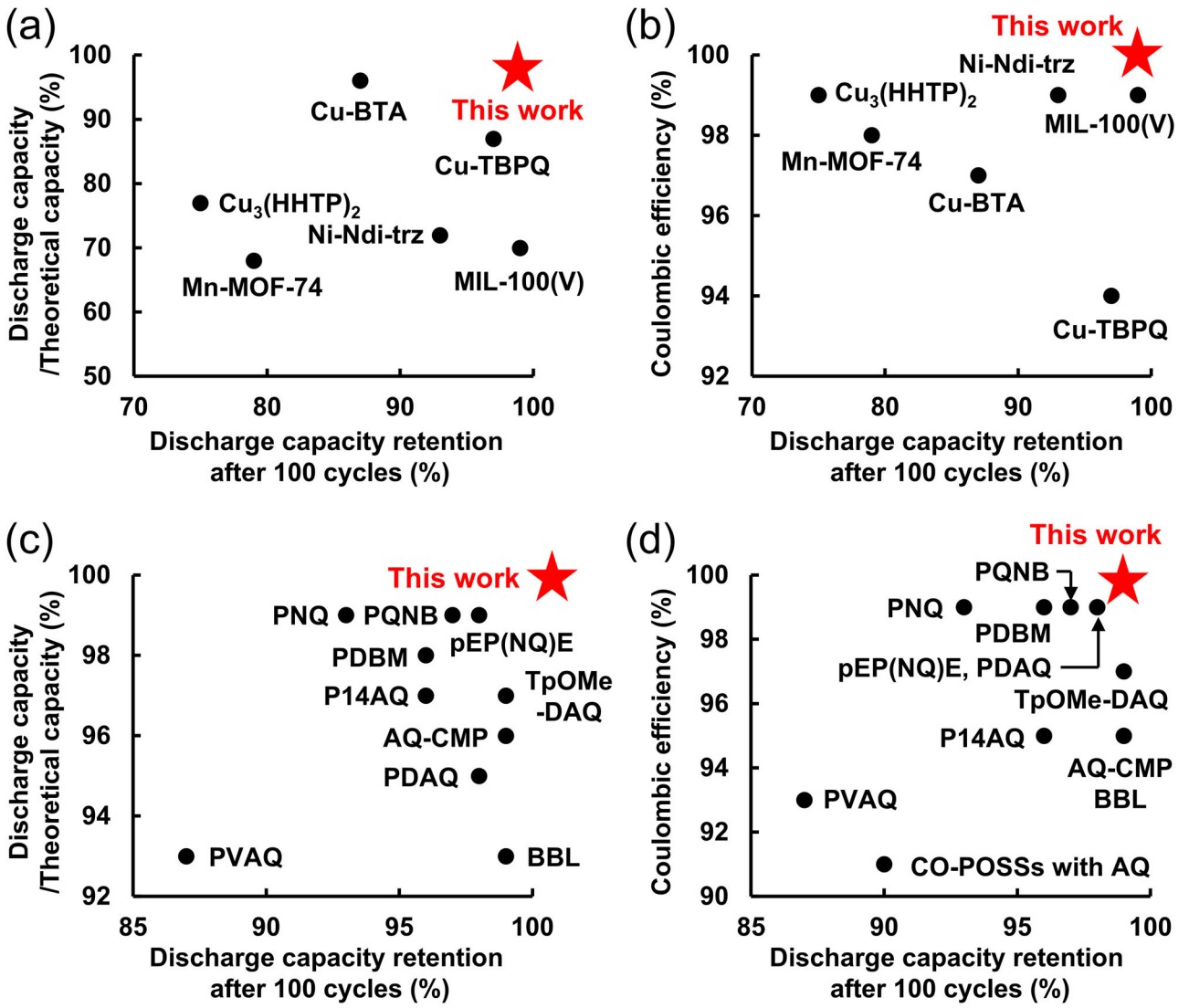

**Fig. 4 | The comparison of battery performance.** The summary of (**a**) discharge capacity/theoretical capacity and discharge capacity retention, **b** Coulombic efficiency and discharge capacity retention of aqueous MOF-based rechargeable batteries[17–22] (further details, see Supplementary Table 3). Although performance between redox-active metal–organic frameworks (RAMOFs) should be compared with half-cell measurements, all previous data were based on full-cell measurements with zinc as the anode. Therefore, in the current work, the performance of RAMOFs was compared with batteries. The summary of (**c**) discharge capacity/theoretical capacity and discharge capacity retention, **d** Coulombic efficiency and discharge capacity retention for aqueous organic–air rechargeable batteries[27–32,38,56,62,68–70] (further details, see Supplementary Table 4).

RAMOFs in aqueous environments. In our continuous work, we will apply the water-resistant RAMOF to the electrode-active material for other aqueous batteries to develop appropriate applications of MOFs. In order to reduce the amount of conductive additives, in our continuous work, we will address the following two strategies. First, we will develop an electrically conductive MOF with high acid resistance for energy storage in strong acidic aqueous electrolytes by focusing on the hard and soft acids and bases principle, which strongly affects the acid stability of coordination bonds in MOFs. Second, we consider the incorporation of in situ-formed conductive polymers in the porous framework, as demonstrated in our previous work using redox-active covalent organic frameworks[31], which could potentially overcome the low intrinsic conductivity of RAMOFs.

## Methods
### Electrode preparation
UiO-66-(OH)$_2$/GMS/PVdF composite electrodes were prepared by drop-casting a slurry of UiO-66-(OH)$_2$, GMS, and PVdF (4:5:1 w/w/w) and $N$-methyl-2-pyrrolidone onto glassy carbon substrates. The mass loading of UiO-66-(OH)$_2$ was adjusted to approximately 0.1–1.0 mg.

### Electrochemical characterization
Electrochemical measurements were performed using a 0.05 M H$_2$SO$_4$ aqueous solution. Cyclic voltammetry and half-cell measurements were conducted using a potentiostat system (HZ-7000, Meiden Hokuto, Japan) comprising a coiled Pt wire as the counter electrode and a RE-1B aqueous reference electrode (Ag/AgCl (3 M NaCl); BAS Inc.) under Ar gas. Half-cell measurement was conducted in the range of from −0.20 to +0.90 V vs. Ag/AgCl.

### MOF–air rechargeable battery evaluation
A tailor-made two-compartment glass cell with an unglazed plate to separate the cathode and anode compartments was employed as the electrochemical cell[27–29]. The UiO-66-(OH)$_2$/GMS/PVdF composite electrode was used as the anode, and a 20% Pt on carbon (Pt/C) paper from Fuel Cell Earth was used as the conventional cathode. Both anode

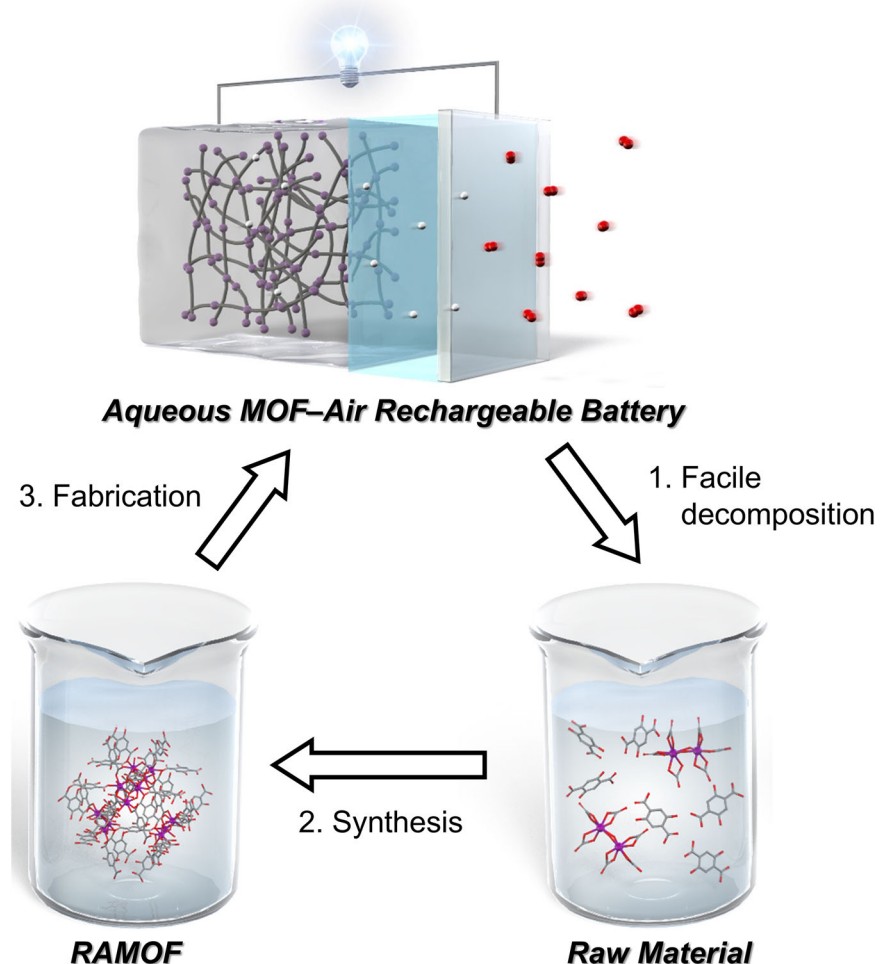

**Fig. 5 | Recycling method for UiO-66-(OH)₂ (RAMOF: redox-active metal–organic framework).** Three-step recycling process: the first step was facile decomposition of the UiO-66-(OH)₂ into its raw materials, the second step was resynthesis of the RAMOF, and the third step was refabrication of the aqueous MOF–air rechargeable battery.

and cathode sections were filled with a 0.05 M $H_2SO_4$ aqueous solution, and the cathode side was open to the air. The battery was evaluated for several cycles to confirm the reproducibility at 20 °C.

## Data availability

The data generated in this study are provided in the Supplementary Information/Source Data file. The data supporting the findings of this study are available within the article and its Supplementary Information. Source data are provided with this paper.

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

## Acknowledgements

This work was partially supported by Grants-in-Aids for Scientific Research from MEXT, Japan, Grant Nos. JP23K17945 (K. Oka), JP23H03827 (K. Oka), JP24K01552 (K. Oka), JP24KJ1576 (R.A.), JP25K21722 (K. Oka). This work was supported by JSPS Bilateral Program Number JPJSBP120258801 (K. Oka). In addition, this work was partially supported by the Environment Research and Technology Development Fund (JPMEERF20241RA4, K. Oka) of the Environmental Restoration and Conservation Agency provided by Ministry of the Environment of Japan. This work was partially supported by the Shorai Foundation for Science and Technology (K. Oka), TEPCO Memorial Foundation (K. Oka), Amano Industry Technology Laboratory (K. Oka), Sugiyama Houkoukai (K. Oka), The Yamada Science Foundation (K. Oka), Kenjiro Takayanagi Foundation (K. Oka), Kansai Research Foundation for Technology Promotion (K. Oka), Yashima Environment Technology Foundation (K. Oka), JACI Prize for Encouraging Young Researcher (K. Oka), Iketani Science and Technology Foundation (K. Oka), Foundation for Interaction in Science & Technology (K. Oka), Ozawa and Yoshikawa Memorial Electronics Research Foundation (K. Oka), Kato Foundation for Promotion of Science (KJ-3416, K. Oka), Hosokawa Powder Technology Foundation (R.A.), and Mishima Kaiun Memorial Foundation (R.A.). This work was partially supported by JST, the establishment of university fellowships towards the creation of science technology innovation (JPMJFS2102, K. Okubo).

## Author contributions

K. Oka conceived and supervised the project. R.A., S.K. and K. Oka designed the research and performed synthesis, characterization, electrochemical experiments, and other major experiments. K. Okubo performed the $N_2$ adsorption measurement. N.S. assisted in FT-IR measurements. H.N. provided GMS to fabricate carbon composite electrodes. R.A. and K. Oka wrote the initial draft of the paper, H.K. provided substantial revisions, and all authors contributed to the editing of the paper.

## Competing interests

The authors declare no competing interests.
