## [Transparent Peer Review file · Nature Communications]

Water-Resistant Redox-Active Metal–Organic Framework

Corresponding Author: Professor Kouki Oka

Version 0:

Reviewer comments:

Reviewer #1

(Remarks to the Author)

Oka et al. proposed a new type of redox-active MOF (RAMOF) and demonstrated its application performance in aqueous MOF-air rechargeable batteries. I do not think the research is innovative enough. Although the author emphasizes that it is a new type of MOF, I still have doubts about the conclusion.

1. The author prepared UiO-66-(OH)₂ with smaller particle size (average particle size: 70±10nm) by reducing the concentration and reducing the reaction time in the microwave. This synthesis method is not very innovative with MOF, but only prepares a smaller particle size of UiO-66-(OH)₂.
2. The rationality of the experimental design. The author only compared the preparation of UiO-66-(OH)₂ with two concentrations and the difference in particle size. Please add a more detailed concentration gradient experiment to prove the relationship between the synthesis method and particle size.
3. The current characterization does not explain why a strong Zr-O bond is obtained. I suspect that it is because of the low energy, short time microwave and low concentration that the Zr-O in ZrOCl₂ is not completely reacted.
4. Supplementary Figure 3. provides a theoretical capacity of 136.8 mAh g⁻¹ for large-sized UiO-66-(OH)₂. However, the theoretical capacity calculated in "Supplementary 2.6 Electrochemical Performance Value Calculation" is 171.9 mAh g⁻¹. Please explain the difference and reason. The main text Figure 3 does not have the theoretical capacity of small-sized UiO-66-(OH)₂. Please add it for comparison.
5. From the application performance of UiO-66-(OH)₂ in aqueous MOF-air rechargeable batteries, it seems to perform well. However, there is no explanation for the essential mechanism of its high performance. Please prove it through some in-situ characterization and DFT calculations.

Reviewer #2

(Remarks to the Author)

The authors reported the synthesis of a water-resistant redox-active metal–organic framework (MOF), UiO-66-(OH)₂, for reversible charge storage in acidic aqueous electrolytes. In addition, the MOF was fabricated into an aqueous MOF–air rechargeable battery, which exhibits high durability and high Coulombic efficiency. This study is interesting for areas of functional porous materials and electrochemical energy storage, which could be suitable for publication in Nat Commun. I would suggest the authors address the comments below to further improve the manuscript:

1. I would suggest the authors investigate the structural changes of the MOF during the electrochemical studies using more detailed in situ or ex situ characterization techniques, in order to further illustrate the mechanism for charge storage.
2. In general, most MOF materials, including the presented RAMOF, show relatively low electrical conductivity, which is a drawback for their electrochemical energy storage application. Could the authors comment on how to address this drawback besides the conventional methods of adding external conductive additives? In addition, could the authors comment on the use of electrically conductive MOFs for similar applications?
3. The authors should provide the full name of GMS in the manuscript.
4. What about the long-term cyclability of the electrode and the battery? For instance, over 1000 cycles?

5. I would suggest the authors investigate the structure and composition of the MOF in the electrode and battery after being charged/discharged for several cycles (e.g. 100 cycles).

Reviewer #3

(Remarks to the Author)

This manuscript (NCOMMS-24-83047-T) presents a RAMOF, UiO-66-(OH)₂, with 1,4-dihydroxybenzene as an organic redox-active material, and discusses its electrochemical performance, including the cycling performance of the assembled aqueous MOF–air rechargeable battery, which is particularly impressive. However, the following issues must be addressed before it can be accepted.

1. The manuscript's logical flow needs to be strengthened. In the Introduction, the authors mention that organic–air rechargeable batteries have an advantage over metal–air rechargeable batteries in terms of cyclability, but they also highlight the issue that organic redox-active materials tend to dissolve or degrade, which results in poor cyclability. Furthermore, the manuscript does not clearly explain why UiO-66-(OH)₂ was chosen as the specific type of RAMOF. The rationale for selecting UiO-66-(OH)₂ as the anode-active material should be clarified in the Introduction.
2. The authors attribute the stable structure and reversible charge/discharge behavior of UiO-66-(OH)₂ to strong Zr-O bonds and a high coordination number. However, there is a lack of direct experimental evidence to support this claim. Spectroscopic data before and after cycling of battery should be provided to substantiate this statement.
3. I recommend that the authors include some in-situ characterization during the battery cycling process to further strengthen the manuscript and enhance its credibility.
4. In Fig. 2(a), the CV curve of the first cycle shows significant differences compared to the subsequent cycles. Please provide an explanation for this observation. Additionally, from the 1st to the 26th cycle, a polarization is observed in the CV curves. Please elaborate on the possible causes of this polarization.
5. The authors tested the EIS of UiO-66-(OH)₂ and reported an ion conductivity of 2.18×10^{-6} S/cm. Besides proton conductivity, do other types of ions contribute to this ion conductivity?
6. Additionally, after the high-rate charging/discharging (e.g., 45 C), does the structure of UiO-66-(OH)₂ undergo any changes? Can the authors provide relevant characterization to confirm its stability, as this would be crucial for demonstrating its high durability and stability.
7. The cycling data of battery presented in the manuscript is impressive, with minimal degradation observed at current rates ranging from 5 C to 45 C. However, how much further can the current rate be increased before noticeable degradation of the battery occurs?

Reviewer #4

(Remarks to the Author)

[Editorial Note: See end of file]

Version 1:

Reviewer comments:

Reviewer #1

(Remarks to the Author)

The authors' response is generally sufficient, reasonable, and scientifically based. By supplementing the paper with experimental data (such as concentration gradient SEM, FT-IR comparisons, in situ/ex situ FT-IR, and DFT calculations) and theoretical analysis, they have enhanced the paper's completeness and persuasiveness, further strengthening its innovativeness and scientific nature. However, some minor issues remain and require further revision:

1. The use of "-" ("-") in various forms throughout the paper makes some data less visually appealing. For example, on Page 12, "-0.1→+0.4 V vs. Ag/AgCl" and Page 12, "-0.1→+0.4 V vs. Ag/AgCl," it is recommended that spaces be added or replaced with "~" to separate word.
2. The order of the supporting figures is confusing. On Page 6, the title of Figure 1 refers to "Supplementary Fig. 5," but Supplementary Fig. 1 is not mentioned until Page 6. Please ensure that all supporting information is included.

Reviewer #2

(Remarks to the Author)

The authors have addressed the reviewers' comments properly; therefore, I suggest acceptance of the manuscript.

Reviewer #3

(Remarks to the Author)

The authors have well addressed the questions, and therefore I recommend it for publication without change.

Reviewer #4

(Remarks to the Author)

The revisions have been thoroughly reviewed and appropriately reflect our suggestions. We are satisfied with the changes and have no further comments.

**Response to the Reviewer 1:**

We appreciate very much your productive comments on our manuscript.

0. Oka *et al.* proposed a new type of redox-active MOF (RAMOF) and demonstrated its application
performance in aqueous MOF-air rechargeable batteries. I do not think the research is innovative
enough. Although the author emphasizes that it is a new type of MOF, I still have doubts about the
conclusion.

Thank you for your comment. Since MOFs are well-known to be structurally unstable in acidic
aqueous solutions owing to their coordination bonds, the application of RAMOF as electrode-active
materials for aqueous batteries has been limited to systems that avoid acidic aqueous electrolytes. By
overcoming this limitation, the current work successfully presented the following three innovative
achievements. First, focusing on the water-resistant MOF with strong Zr–O bonds and a large
coordination number, we demonstrated a structurally stable RAMOF in an acidic aqueous electrolyte.
We achieved reversible charge storage capabilities with nearly the theoretical capacity of the
RAMOF, for the first time, owing to its optimized particle size, high porosity (BET surface area:
$1075 \text{ m}^2 \text{ g}^{-1}$, pore size: 0.62 nm), and proton conductivity ($2.18 \times 10^{-6} \text{ S cm}^{-1}$ under 95% RH at 30°C).
We also demonstrated high Coulombic efficiency, owing to the high crystallinity of the RAMOF and
its proton conductivity which enabled efficient ion transport and uniform reactions throughout the
material. Second, by using this RAMOF as an anode-active material, an aqueous MOF–air
rechargeable battery achieved reversible charge storage capabilities with nearly the theoretical
capacity, high durability, and high Coulombic efficiency as summarized in Supplementary Tables 3
and 4. Third, we demonstrated the material recycling of the RAMOF through a simple treatment with
an aqueous carbonate solution because the coordination bonds of the RAMOF exhibited instability
in aqueous carbonate solutions while retaining robustness in acidic aqueous solutions. The current
work suggests that RAMOFs can overcome long-standing limitations in aqueous applications and
exhibit inherent advantages as electrode-active materials.

Therefore, we have added explanations to clarify the significance of the current work on page 5,
lines 7–19, and page 24, lines 2–5, as below.

“By introducing redox-active *p*-hydroquinone units (redox potential: approximately +0.1 V vs.
Ag/AgCl^{27,36}), in place of benzene in the organic linker of the acid-resistant MOF, we prepare the
acid-resistant RAMOF UiO-66-(OH)₂, which achieves reversible charge storage with an ideal
capacity close to the theoretical capacity based on the molecular weight even in acidic aqueous
electrolytes, owing to its optimized particle size, high porosity, and proton conductivity. In addition,
the RAMOF exhibits high durability and high Coulombic efficiency owing to its high crystallinity
and proton conductivity. Then, by using the RAMOF as an anode-active material, an aqueous MOF–
air rechargeable battery is fabricated. In addition, after use of the battery, we recycle the RAMOF
through a simple treatment with an aqueous carbonate solution because the coordination bonds of the
RAMOF exhibit instability in aqueous carbonate solutions while retaining robustness in acidic
aqueous solutions.”

“Since MOFs are well-known to be structurally unstable in acidic aqueous solutions, owing to their
coordination bonds, the application of RAMOF as electrode-active materials for aqueous batteries
has been limited to systems that avoid acidic aqueous electrolytes.”

1. The author prepared UiO-66-(OH)₂ with smaller particle size (average particle size: 70±10nm) by
reducing the concentration and reducing the reaction time in the microwave. This synthesis method is
not very innovative with MOF, but only prepares a smaller particle size of UiO-66-(OH)₂.

Thank you for your comment. As mentioned above in response to your Comment 0, one of the

novelties of the current work lies in achieving reversible charge storage with nearly the theoretical
capacity of the RAMOF by optimizing its particle size. As described in reference 11 in the revised
supplementary information, we controlled the particle size of UiO-66-(OH)₂ (average particle size:
70 ± 10 nm) by reducing the reaction concentration and the reaction time. Despite the inherently low
electrical conductivity of UiO-66-(OH)₂ (3.81×10⁻⁸ S cm⁻¹ under 95% RH at 30°C), this work
achieved reversible charge storage with nearly the theoretical capacity for the first time in acidic
aqueous electrolytes. This work demonstrates the practical design significance of particle size control
in enabling the reversible charge storage of RAMOFs.

Therefore, we have added the novelty of the material preparation in the current work on page 24,
lines 7–11, as below.

“The UiO-66-(OH)₂ was structurally stable even in acidic aqueous electrolytes owing to its strong
Zr–O bonds and the largest coordination number in MOFs, and achieved reversible charge storage
with an ideal capacity close to the theoretical capacity of the UiO-66-(OH)₂ for the first time by
reducing its particle size.”

2. The rationality of the experimental design. The author only compared the preparation of UiO-66-
(OH)₂ with two concentrations and the difference in particle size. Please add a more detailed
concentration gradient experiment to prove the relationship between the synthesis method and particle
size.

Thank you for your comment. According to your comment, we conducted additional experiments
to investigate the relationship between precursors’ concentrations and particle sizes. UiO-66-(OH)₂
was synthesized under a series of different precursors’ concentrations, and SEM was used to evaluate
the resulting particle sizes. As shown in Supplementary Fig. 1, smaller particle sizes were obtained
at lower precursors’ concentrations. This trend demonstrated a positive correlation between
precursors’ concentrations and particle sizes under identical reaction conditions, indicating that
particle sizes could be tuned in a precursors’ concentration-dependent manner. These results
supported the rationality of our experimental design and the validity of particle size control through
precursors’ concentrations.

We have presented the precursors’ concentration-dependent changes in particle sizes in
Supplementary Fig. 1 and added its explanation on page 7, lines 11–14, as below.

“As shown in Supplementary Fig. 1, smaller particle sizes were obtained at lower precursors’
concentrations. This trend demonstrated a positive correlation between precursors’ concentrations
and particle sizes under otherwise identical reaction conditions.”

**Supplementary Figure 1. Characterization of UiO-66-(OH)₂.** SEM images of UiO-66-(OH)₂
prepared at different precursors' concentrations. In each case, concentrations of ZrOCl₂·8H₂O and
2,5-dihydroxyterephthalic acid were used: (a) 50 mM, (b) 100 mM, and (c) 200 mM, respectively.
The other reaction conditions were identical to those described in the Experimental Section 2.1. (d)
The effect of concentrations of ZrOCl₂·8H₂O and 2,5-dihydroxyterephthalic acid on the average
particle size.

3. The current characterization does not explain why a strong Zr-O bond is obtained. I suspect that it
is because of the low energy, short time microwave and low concentration that the Zr-O in ZrOCl₂ is
not completely reacted.

Thank you for your comment. ZrOCl₂·8H₂O had a Zr=O bond in its molecular structure, not a Zr-
O bond. As shown in FT-IR spectra in Supplementary Fig. 6, ZrOCl₂ was almost completely reacted
after the preparation reaction of UiO-66-(OH)₂ in the current work, because the peak at 1023 cm⁻¹,
derived from Zr=O in ZrOCl₂·8H₂O, disappeared in UiO-66-(OH)₂.

We hypothesized that if low temperature, short reaction time, and low precursors' concentrations
affected the strength of Zr-O bonds, then UiO-66-(OH)₂ synthesized under harsher conditions,
compared with UiO-66-(OH)₂ synthesized under the reaction conditions in the Experimental Section
2.1 (our sample), would exhibit changes in the FT-IR peak positions and intensities corresponding to
O-Zr-O and Zr-(OC) which were all kinds of Zr-O bonds contained in UiO-66-(OH)₂. In order to
support that strong Zr-O bonds were formed under the reaction conditions described in the
Experimental Section 2.1, we also synthesized UiO-66-(OH)₂ under harsher conditions (higher
temperature, longer reaction time, and higher precursors' concentration, where the details are
provided in the caption of Supplementary Fig. 6) than those previously reported (reference 40 in the
revised manuscript), and measured FT-IR spectra (Supplementary Fig. 6). As shown in
Supplementary Fig. 6, the characteristic peaks corresponding to O-Zr-O and Zr-(OC) bonds
appeared at 662 cm⁻¹ and 575 cm⁻¹, respectively, for both samples. Furthermore, the intensities of

these two peaks were identical. These results indicated that strong O–Zr–O and Zr–(OC) bonds were
successfully formed, even under the reaction conditions (Experimental Section 2.1), to resist acidic
aqueous solutions.

We have added these FT-IR spectra as Supplementary Fig. 6 and the explanation on page 8, lines
10–19, as below.

“In order to support that strong Zr–O bonds were formed, we also synthesized UiO-66-(OH)₂ under
harsher conditions (higher temperature, longer reaction time, and higher precursors’ concentration,
where the details are provided in the caption of Supplementary Fig. 6) than those previously
reported⁴⁰, and measured Fourier-transform infrared (FT-IR) spectra (Supplementary Fig. 6). As
shown in Supplementary Fig. 6, the peak positions derived from O–Zr–O and Zr–(OC) in UiO-66-
(OH)₂ prepared under different reaction conditions were identical (662 cm⁻¹⁴⁵ and 575 cm⁻¹⁴²,
respectively), indicating that strong Zr–O bonds were successfully formed, even under the reaction
conditions (Experimental Section 2.1), to resist acidic aqueous solutions.”

**Supplementary Figure 6. Characterization of UiO-66-(OH)₂.** FT-IR spectra of UiO-66-(OH)₂
prepared with 95°C, reaction time of 15 min, and lower precursors’ concentrations of 50 mM (black,
Supplementary Table 1 Entry 2), UiO-66-(OH)₂ prepared with higher temperature (110°C), longer
reaction time in the microwave (60 min), and higher precursors’ concentrations (200 mM) (red), and
ZrOCl₂·8H₂O (blue). By comparing blue and black lines, ZrOCl₂ was almost completely reacted after
the preparation reaction of UiO-66-(OH)₂, because the peak at 1023 cm⁻¹, derived from Zr=O in
ZrOCl₂·8H₂O¹², disappeared in UiO-66-(OH)₂. Inset: The spectra (black and red) were baseline-
corrected by specifying three points at 520, 940, and 1280 cm⁻¹ and subtracting the linear baseline
defined by these points^{13,14} because no peaks were observed in these points. After that, the spectra
(black and red) were normalized by the peaks at 1235 cm⁻¹, attributed to the C–(OH) of the organic
linkers¹⁵⁻¹⁷, which remained unchanged with respect to O–Zr–O and Zr–(OC).

4-1. Supplementary Figure 3. provides a theoretical capacity of 136.8 mAh g⁻¹ for large-sized UiO-
66-(OH)₂. However, the theoretical capacity calculated in “Supplementary 2.6 Electrochemical
Performance Value Calculation” is 171.9 mAh g⁻¹. Please explain the difference and reason.

Thank you for your comment. The difference in the theoretical capacities between large- (136.8
mAh g⁻¹) and small-sized UiO-66-(OH)₂ (171.9 mAh g⁻¹) was caused by the number of missing

organic linkers. The detailed calculation method of the number of missing organic linkers was
described in the Experimental Section 2.3 and the caption of Supplementary Fig. 3. According to a
previous report (reference 11 in the revised supplementary information), short reaction time (*e.g.*, our
reaction conditions described in the Experimental Section 2.1) decreased the particle size and the
number of missing organic linkers. This was likely because a short reaction time suppressed the
thermodynamically favorable formation of defects by limiting the competition of acetic acid with the
organic linker for coordination sites on Zr clusters, resulting in a less defective structure.

Therefore, we have added thermogravimetric analysis of UiO-66-(OH)₂ with a large particle size
(Supplementary Fig. 3) to the supplementary information and the explanation of the calculation of
the theoretical capacity in the caption of Supplementary Fig. 3, as below.

**Supplementary Figure 3. Thermogravimetric analysis of UiO-66-(OH)₂ with large particle size**
**under air.** Theoretically, the normalized weight percentage at 280°C was 246.2%, which led to a
24.37% weight loss per organic linker. Experimentally, the normalized weight percentage at 280°C
was 197.7%. Therefore, the number of organic linkers containing UiO-66-(OH)₂ was calculated to
be 4.01, *i.e.*, 1.99 organic linker defects per Zr cluster. Since organic linker defect sites were occupied
by water and hydroxide anions, the molecular weight of UiO-66-(OH)₂ was 1571.3 g mol⁻¹
(Zr₆O₄(OH)₄(C₈H₄O₆)_{4.01}(H₂O)_{3.98}(OH)_{1.99})⁷. Its theoretical capacity was calculated as 136.8 mAh
g⁻¹ according to the formula given in the Experimental Section 2.6. According to a previous report,
short reaction time decreased the particle size and the number of missing organic linkers¹¹. This was
likely because a short reaction time suppressed the thermodynamically favorable formation of defects
by limiting the competition with acetic acid, resulting in a less defective structure.

4-2. The main text Figure 3 does not have the theoretical capacity of small-sized UiO-66-(OH)₂. Please
add it for comparison.

Thank you for your comment. For clarity, we have added a dotted line indicating the theoretical
capacity to Fig. 3c and the explanation of the dotted line in the caption on page 19, lines 3–4, as below.

**Fig. 3** (c) Charging (black)/discharging (red) curves of the aqueous MOF–air rechargeable battery at
5 C. The dotted line represents the theoretical capacity based on the molecular weight of UiO-66-
(OH)₂ (171.9 mAh g⁻¹). Inset: The battery cycle test (14 C). At 14 C, the battery achieved a discharge
capacity of more than 98% of the theoretical capacity based on the molecular weight of UiO-66-
(OH)₂. Therefore, we performed a cycling test at 14 C.

5. From the application performance of UiO-66-(OH)₂ in aqueous MOF-air rechargeable batteries, it
seems to perform well. However, there is no explanation for the essential mechanism of its high
performance. Please prove it through some *in-situ* characterization and DFT calculations.

Thank you for your comment. In order to prove the essential mechanism of the high performance
of UiO-66-(OH)₂, we performed *in situ* FT-IR analysis in addition to *ex situ* FT-IR analysis. Since
the electrochemical behavior of UiO-66-(OH)₂ during the charge/discharge process was identical in
both half-cell and battery, we conducted *ex situ* and *in situ* FT-IR measurements using the electrode
in the half-cell. By focusing on the initial state of the structure of UiO-66-(OH)₂, as shown in
Supplementary Fig. 11 (DFT calculations), we found that the optimized structure of the cluster of
UiO-66-(OH)₂ had hydrogen bonds between hydroxyl and carboxy groups. Upon oxidation at around
+0.7 V vs. Ag/AgCl (Supplementary Fig. 9), the formation of the C=O was confirmed by *ex situ* FT-
IR analysis. Based on *in situ* FT-IR analysis (Supplementary Fig. 10) and previous literature (redox
reactions of non-hydrogen-bonded *p*-hydroquinone (references 47 and 48 in the revised manuscript)),
the oxidation peaks at around +0.3 V and +0.7 V vs. Ag/AgCl were attributed to the oxidation of
non-hydrogen-bonded and hydrogen-bonded C–(OH) of *p*-hydroquinone, respectively. Furthermore,
we conducted DFT calculations to identify redox sites of the oxidation state of UiO-66-(OH)₂,
following the methodologies reported in previous works (references 56 and 57 in the revised
manuscript). The molecular electrostatic potential (MESP) mapping revealed that oxygen atoms of
the *p*-benzoquinone moiety exhibited a significant negative MESP value, indicating their suitability
for proton storage. Based on *ex situ* and *in situ* FT-IR analyses and DFT calculations, we confirmed
the redox reaction between *p*-hydroquinone and *p*-benzoquinone in the UiO-66-(OH)₂.

We have added the result of *in situ* FT-IR analysis as Supplementary Fig. 10, the result of
calculations as Supplementary Figs. 11 and 12, and their explanations on page 12, line 1 – page 14,
line 6, as below.

“In order to investigate the details of the charge storage mechanism of UiO-66-(OH)₂, as shown in
Supplementary Figs. 9 and 10, we performed *ex situ* and *in situ* FT-IR analyses. By focusing on the
initial state of the structure of UiO-66-(OH)₂, as shown in Supplementary Fig. 11 and Supplementary
Table 5, we found that the density functional theory (DFT)-optimized structure of the cluster of UiO-
66-(OH)₂ exhibited an O···O distance of 2.48 Å⁴⁹, suggesting that the initial state of the structure of

UiO-66-(OH)₂ had hydrogen bonds between the protons of C-(OH) of *p*-hydroquinone and carboxy
groups. As shown in Fig. 2a and Supplementary Fig. 9a, an irreversible oxidation peak appeared at
around +0.7 V vs. Ag/AgCl upon sweeping the potential in the positive direction from +0.50 V to
+0.90 V vs. Ag/AgCl. As shown in Supplementary Figs. 9a and 9b, in the *ex situ* FT-IR spectrum
after applying the potential at +0.90 V vs. Ag/AgCl, a new peak appeared at 1638 cm⁻¹, derived from
C=O⁵⁰, which indicated that the new peak was attributed to the formation of C=O by the oxidation
(Supplementary Fig. 9c). This attribution was further supported by the finding that the oxidation
potential shift to the positive direction was presumably caused by hydrogen-bond formation, as
reported in previous works^{51,52}. In addition, as shown in Supplementary Fig. 10b, the peak intensity
of 1235 cm⁻¹ at the initial state of the red solid-line spectrum, attributed to C-(OH) of *p*-
hydroquinone⁵³⁻⁵⁵, decreased upon oxidation, resulting in the purple solid-line spectrum, thereby
suggesting the oxidation of hydrogen bonded C-(OH) of *p*-hydroquinone. From the above results,
as shown in Supplementary Fig. 10c, the oxidation peak at around +0.7 V vs. Ag/AgCl was attributed
to the oxidation of hydrogen-bonded C-(OH) of *p*-hydroquinone to C=O of *p*-benzoquinone. After
that, as shown in Supplementary Fig. 10a, upon sweeping the potential from +0.90 V to -0.20 V vs.
Ag/AgCl, we observed a reduction peak at around 0.0 V vs. Ag/AgCl and an increase of the peak
intensity from the purple solid- to the blue solid-line spectra (Supplementary Fig. 10b). As shown in
Supplementary Fig. 12 and Supplementary Table 6, the molecular electrostatic potential (MESP)
map suggested that C=O of *p*-benzoquinone in the oxidation state of UiO-66-(OH)₂ would be the
reduction site for proton storage owing to the strongly negative MESP value of the oxygen atoms in
*p*-benzoquinone^{56,57}. These results indicated that the reduction peak around 0.0 V vs. Ag/AgCl
(Supplementary Fig. 10b) was attributed to the reduction of C=O of *p*-benzoquinone to C-(OH) of
*p*-hydroquinone (Supplementary Fig. 10c). After that, upon sweeping the potential from -0.20 V to
+0.90 V vs. Ag/AgCl (Supplementary Fig. 10a), as shown in Supplementary Fig. 10b, we observed
two kinds of oxidation peaks at around +0.3 V (polarized) and +0.7 V vs. Ag/AgCl, and observed a
decrease in peak intensity from the blue solid- via red dotted- to purple dotted-line spectra. Based on
these peak intensity changes (Supplementary Figs. 10b and 10c) and previous literature (redox
reactions of non-hydrogen-bonded *p*-hydroquinone^{47,48}), the oxidation peaks around +0.3 V and +0.7
V vs. Ag/AgCl were attributed to non-hydrogen-bonded and hydrogen-bonded C-(OH) of *p*-
hydroquinone, respectively. From the above analyses, as shown in Fig. 2a, the oxidation peak at
around +0.7 V vs. Ag/AgCl was attributed to the oxidation of hydrogen-bonded C-(OH) of *p*-
hydroquinone (Supplementary Fig. 10c), and the redox peak in the range of -0.1–+0.4 V vs.
Ag/AgCl was attributed to the redox reaction of non-hydrogen-bonded C-(OH) of *p*-hydroquinone
(Supplementary Fig. 10c).”

 **Supplementary Figure 9. Identification of conformationally changed sites in UiO-66-(OH)₂.** (a,
 b) *Ex situ* FT-IR spectra of UiO-66-(OH)₂ (black) and UiO-66-(OH)₂ after applying a potential of
 +0.90 V vs. Ag/AgCl for 2 h in a 0.05 M H₂SO₄ aqueous solution. This spectrum was obtained by
 converting the transmittance data using the following equation. “A = log (1/T) (A: Absorbance, T:
 Transmittance)” Potentiostatic electrolysis was performed using a UiO-66-(OH)₂ electrode, which
 was fabricated by drop-casting a slurry of UiO-66-(OH)₂ and *N*-methyl-2-pyrrolidone onto a glassy
 carbon electrode and drying at 120°C for 2 h. After performing potentiostatic electrolysis, the
 electrode was immersed in water to remove the electrolyte for measuring the *ex situ* FT-IR spectrum.
 The spectra were normalized by the peaks at 1590 cm⁻¹, attributed to the COO⁻ of the organic
 linkers¹⁶, which remained unchanged before and after the reaction. (c) Molecular structural changes
 estimated from the difference of *ex situ* FT-IR spectra.

 **Supplementary Figure 10. Identification of conformationally changed sites in UiO-66-(OH)₂**
 **based on *in situ* measurement.** (a) Different charge/discharge states selected from the cyclic
 voltammogram and (b) corresponding *in situ* FT-IR spectra. The measurement was performed using
 an electrode composed of UiO-66-(OH)₂ and SWNT (5:1 w/w), in which the carbon ratio is smaller
 than that described in the Methods. Therefore, a part of the C-(OH) groups of *p*-hydroquinone
 contributed to the redox reaction. Since a strong peak of H₂O was observed at around 1600 cm⁻¹, we
 focused on the peak at around 1235 cm⁻¹, corresponding to the C-(OH) group of *p*-hydroquinone¹⁵⁻
 ¹⁷. *In situ* FT-IR spectra were baseline-corrected by specifying two points at 1210 and 1260 cm⁻¹ and
 subtracting the linear baseline defined by these points¹³⁻¹⁴. (c) The estimated structural changes during
 charge/discharge measurements. Once the electrode was oxidized, the redox peak in the range of
 12 -0.1–+0.4 V vs. Ag/AgCl became predominant compared to the irreversible oxidation peak at +0.7
 V vs. Ag/AgCl (Fig. 2a), presumably because it took time for the hydrogen bonds to form¹⁸.

 **Supplementary Figure 11. The optimized structure of the cluster of UiO-66-(OH)₂.** The O...O
 distance was calculated to be 2.48 Å, which indicated the formation of hydrogen bonds between

hydroxyl and carboxy groups¹⁹.

**Supplementary Figure 12. Identification of redox sites in the oxidation state of UiO-66-(OH)₂**
**based on MESP analysis.** On the van der Waals surface of the atoms, the red and blue sites are the
electron-rich and electron-deficient regions, respectively. Since it was difficult to take into account
whether the hydroxyl groups formed hydrogen bonds with carboxy groups in UiO-66-(OH)₂, the
calculation was performed for the cluster of the oxidation state of UiO-66-(OH)₂ with *p*-
benzoquinone as an organic linker.

Based on the above, *ex situ* and *in situ* FT-IR analyses, as well as DFT calculations, confirmed the
essential mechanism of its high-performance RAMOF, owing to the reversible charge storage
capabilities of the *p*-hydroquinone moiety in the RAMOF.

Therefore, we have added the explanation on page 24, lines 5–7, as below.

“In the current work, we demonstrated a high-performance RAMOF with *p*-hydroquinone as an
organic linker, whose reversible charge storage capability was elucidated through *ex situ* and *in situ*
FT-IR analyses and DFT calculations.”

Thank you very much again for your productive comments. We hope that our replies are acceptable
to you.

**Response to the Reviewer 2:**

We appreciate very much your positive comments on our manuscript.

1. I would suggest the authors investigate the structural changes of the MOF during the
electrochemical studies using more detailed *in situ* or *ex situ* characterization techniques, in order to
further illustrate the mechanism for charge storage.

Thank you for your comment. In order to further illustrate the charge storage mechanism, we
performed *in situ* FT-IR analysis in addition to *ex situ* FT-IR analysis. Since the electrochemical
behavior of UiO-66-(OH)₂ during the charge/discharge process was identical in both half-cell and
battery, we conducted *ex situ* and *in situ* FT-IR measurements using the electrode in the half-cell. By
focusing on the initial state of the structure of UiO-66-(OH)₂, as shown in Supplementary Fig. 11
(DFT calculations), we found that the optimized structure of the cluster of UiO-66-(OH)₂ had
hydrogen bonds between hydroxyl and carboxy groups. Upon oxidation at around +0.7 V vs.
Ag/AgCl (Supplementary Fig. 9), the formation of the C=O was confirmed by *ex situ* FT-IR analysis.
Based on *in situ* FT-IR analysis (Supplementary Fig. 10) and previous literature (redox reactions of
non-hydrogen-bonded *p*-hydroquinone (references 47 and 48 in the revised manuscript)), the
oxidation peaks at around +0.3 V and +0.7 V vs. Ag/AgCl were attributed to the oxidation of non-
hydrogen-bonded and hydrogen-bonded C-(OH) of *p*-hydroquinone, respectively. Furthermore, we
conducted DFT calculations to identify redox sites of the oxidation state of UiO-66-(OH)₂, following
the methodologies reported in previous works (references 56 and 57 in the revised manuscript). The
molecular electrostatic potential (MESP) mapping revealed that oxygen atoms of the *p*-benzoquinone
moiety exhibited a significant negative MESP value, indicating their suitability for proton storage.
Based on *ex situ* and *in situ* FT-IR analyses and DFT calculations, we confirmed the redox reaction
between *p*-hydroquinone and *p*-benzoquinone in the UiO-66-(OH)₂.

We have added the result of measurements as Supplementary Fig. 10 and its explanation on page
12, line 1 – page 14, line 6, as below.

“In order to investigate the details of the charge storage mechanism of UiO-66-(OH)₂, as shown in
Supplementary Figs. 9 and 10, we performed *ex situ* and *in situ* FT-IR analyses. By focusing on the
initial state of the structure of UiO-66-(OH)₂, as shown in Supplementary Fig. 11 and Supplementary
Table 5, we found that the density functional theory (DFT)-optimized structure of the cluster of UiO-
66-(OH)₂ exhibited an O···O distance of 2.48 Å⁴⁹, suggesting that the initial state of the structure of
UiO-66-(OH)₂ had hydrogen bonds between the protons of C-(OH) of *p*-hydroquinone and carboxy
groups. As shown in Fig. 2a and Supplementary Fig. 9a, an irreversible oxidation peak appeared at
around +0.7 V vs. Ag/AgCl upon sweeping the potential in the positive direction from +0.50 V to
+0.90 V vs. Ag/AgCl. As shown in Supplementary Figs. 9a and 9b, in the *ex situ* FT-IR spectrum
after applying the potential at +0.90 V vs. Ag/AgCl, a new peak appeared at 1638 cm⁻¹, derived from
C=O⁵⁰, which indicated that the new peak was attributed to the formation of C=O by the oxidation
(Supplementary Fig. 9c). This attribution was further supported by the finding that the oxidation
potential shift to the positive direction was presumably caused by hydrogen-bond formation, as
reported in previous works^{51,52}. In addition, as shown in Supplementary Fig. 10b, the peak intensity
of 1235 cm⁻¹ at the initial state of the red solid-line spectrum, attributed to C-(OH) of *p*-
hydroquinone⁵³⁻⁵⁵, decreased upon oxidation, resulting in the purple solid-line spectrum, thereby
suggesting the oxidation of hydrogen bonded C-(OH) of *p*-hydroquinone. From the above results,
as shown in Supplementary Fig. 10c, the oxidation peak at around +0.7 V vs. Ag/AgCl was attributed
to the oxidation of hydrogen-bonded C-(OH) of *p*-hydroquinone to C=O of *p*-benzoquinone. After
that, as shown in Supplementary Fig. 10a, upon sweeping the potential from +0.90 V to -0.20 V vs.
Ag/AgCl, we observed a reduction peak at around 0.0 V vs. Ag/AgCl and an increase of the peak

intensity from the purple solid- to the blue solid-line spectra (Supplementary Fig. 10b). As shown in
 Supplementary Fig. 12 and Supplementary Table 6, the molecular electrostatic potential (MESP)
 map suggested that C=O of *p*-benzoquinone in the oxidation state of UiO-66-(OH)₂ would be the
 reduction site for proton storage owing to the strongly negative MESP value of the oxygen atoms in
 *p*-benzoquinone^{56,57}. These results indicated that the reduction peak around 0.0 V vs. Ag/AgCl
 (Supplementary Fig. 10b) was attributed to the reduction of C=O of *p*-benzoquinone to C-(OH) of
 *p*-hydroquinone (Supplementary Fig. 10c). After that, upon sweeping the potential from -0.20 V to
 +0.90 V vs. Ag/AgCl (Supplementary Fig. 10a), as shown in Supplementary Fig. 10b, we observed
 two kinds of oxidation peaks at around +0.3 V (polarized) and +0.7 V vs. Ag/AgCl, and observed a
 decrease in peak intensity from the blue solid- via red dotted- to purple dotted-line spectra. Based on
 these peak intensity changes (Supplementary Figs. 10b and 10c) and previous literature (redox
 reactions of non-hydrogen-bonded *p*-hydroquinone^{47,48}), the oxidation peaks around +0.3 V and +0.7
 V vs. Ag/AgCl were attributed to non-hydrogen-bonded and hydrogen-bonded C-(OH) of *p*-
 hydroquinone, respectively. From the above analyses, as shown in Fig. 2a, the oxidation peak at
 around +0.7 V vs. Ag/AgCl was attributed to the oxidation of hydrogen-bonded C-(OH) of *p*-
 hydroquinone (Supplementary Fig. 10c), and the redox peak in the range of -0.1–+0.4 V vs.
 Ag/AgCl was attributed to the redox reaction of non-hydrogen-bonded C-(OH) of *p*-hydroquinone
 (Supplementary Fig. 10c).”

 **Supplementary Figure 9. Identification of conformationally changed sites in UiO-66-(OH)₂.** (a,
 b) *Ex situ* FT-IR spectra of UiO-66-(OH)₂ (black) and UiO-66-(OH)₂ after applying a potential of

+0.90 V vs. Ag/AgCl for 2 h in a 0.05 M H₂SO₄ aqueous solution. This spectrum was obtained by
 converting the transmittance data using the following equation. “ $A = \log(1/T)$ (A : Absorbance, T :
 Transmittance)” Potentiostatic electrolysis was performed using a UiO-66-(OH)₂ electrode, which
 was fabricated by drop-casting a slurry of UiO-66-(OH)₂ and *N*-methyl-2-pyrrolidone onto a glassy
 carbon electrode and drying at 120°C for 2 h. After performing potentiostatic electrolysis, the
 electrode was immersed in water to remove the electrolyte for measuring the *ex situ* FT-IR spectrum.
 The spectra were normalized by the peaks at 1590 cm⁻¹, attributed to the COO⁻ of the organic
 linkers¹⁶, which remained unchanged before and after the reaction. (c) Molecular structural changes
 estimated from the difference of *ex situ* FT-IR spectra.

**Supplementary Figure 10. Identification of conformationally changed sites in UiO-66-(OH)₂**
 **based on *in situ* measurement.** (a) Different charge/discharge states selected from the cyclic
 voltammogram and (b) corresponding *in situ* FT-IR spectra. The measurement was performed using
 an electrode composed of UiO-66-(OH)₂ and SWNT (5:1 w/w), in which the carbon ratio is smaller
 than that described in the Methods. Therefore, a part of the C-(OH) groups of *p*-hydroquinone
 contributed to the redox reaction. Since a strong peak of H₂O was observed at around 1600 cm⁻¹, we
 focused on the peak at around 1235 cm⁻¹, corresponding to the C-(OH) group of *p*-hydroquinone¹⁵⁻
 ¹⁷. *In situ* FT-IR spectra were baseline-corrected by specifying two points at 1210 and 1260 cm⁻¹ and
 subtracting the linear baseline defined by these points¹³⁻¹⁴. (c) The estimated structural changes during
 charge/discharge measurements. Once the electrode was oxidized, the redox peak in the range of
 22 -0.1→+0.4 V vs. Ag/AgCl became predominant compared to the irreversible oxidation peak at +0.7
 V vs. Ag/AgCl (Fig. 2a), presumably because it took time for the hydrogen bonds to form¹⁸.

2. In general, most MOF materials, including the presented RAMOF, show relatively low electrical
 conductivity, which is a drawback for their electrochemical energy storage application. Could the
 authors comment on how to address this drawback besides the conventional methods of adding
 external conductive additives? In addition, could the authors comment on the use of electrically
 conductive MOFs for similar applications?

Thank you for your comment. We also recognize that the relatively low electrical conductivity of

1 most MOF materials, including the RAMOF presented in the current work, is a significant limitation
for their widespread use in electrochemical energy storage applications. In the current work, we
addressed this drawback by reducing the particle size of UiO-66-(OH)₂, which shortened the ion and
electron transport pathways and improved redox accessibility, thereby enhancing electrochemical
performance without relying solely on excessive amounts of conductive additives.

As another possible approach to address the low electrical conductivity of most RAMOF materials,
we previously demonstrated the incorporation of *in situ*-formed conductive polymers within redox-
active covalent organic frameworks, resulting in effective electron conduction throughout the
materials (reference 31 in the revised manuscript). Applying a similar approach to the RAMOF in
the current work could potentially overcome its low electrical conductivity by enabling electrical
conduction throughout the material.

Electrically conductive MOFs are indeed promising materials for electrochemical applications
because they can reduce or eliminate the requirement for conductive additives (references 17, 20, and
22 in the revised manuscript). Significant progress has been made in this area, including the important
contribution by C. Xu *et al.* (reference 6 in the revised manuscript), who demonstrated charge storage
capabilities of electrically conductive MOFs in aqueous electrolytes across a wide pH range from 3
to 11. In our continuous work, by optimizing metal–linker combinations from the perspective of the
hard and soft acids and bases (HSAB) principle, we would develop acid-resistant and electrically
conductive MOFs that maintain their crystallinity, exhibiting outstanding charge storage capabilities
(*e.g.*, durability) even in strongly acidic aqueous electrolytes (pH < 2).

Accordingly, we have added the explanation of the continuous work on page 25, lines 3–10, as
below.

“In order to reduce the amount of conductive additives, in our continuous work, we will address the
following two strategies. First, we will develop an electrically conductive MOF with high acid
resistance for energy storage in strong acidic aqueous electrolytes by focusing on the hard and soft
acids and bases principle, which strongly affects the acid stability of coordination bonds in MOFs.
Second, we consider the incorporation of *in situ*-formed conductive polymers in the porous
framework, as demonstrated in our previous work using redox-active covalent organic frameworks³¹,
which could potentially overcome the low intrinsic conductivity of RAMOFs.”

3. The authors should provide the full name of GMS in the manuscript.

Thank you very much for pointing out our mistake. According to your comment, we have added
the full name of GMS on page 15, lines 17–19, as below.

“Then, we evaluated the redox capability of the UiO-66-(OH)₂/graphene mesosponge
(GMS)/PVdF composite electrode in a 0.05 M H₂SO₄ aqueous solution.”

4. What about the long-term cyclability of the electrode and the battery? For instance, over 1000
cycles?

Thank you for your comment. According to your comment, we conducted a long-term cycle test of
the electrode (1000 cycles). As shown in Supplementary Fig. 20, the electrode retained more than
95% of its initial capacity even after 1000 cycles, indicating high cyclability of the RAMOF even in
an acidic aqueous electrolyte.

We have included the result of the long-term cycle test of the electrode as Supplementary Fig. 20
and its explanation on page 17, lines 2–5, as below.

“As shown in Supplementary Fig. 20, a long-term cycle test of the electrode was also performed.
The UiO-66-(OH)₂/GMS/PVdF composite electrode retained over 95% of its initial capacity even
after 1000 cycles, demonstrating its high cyclability.”

**Supplementary Figure 20. The long-term cycle test of the electrode at 45 C.**

In addition, according to your comment, we also conducted a long-term battery cycle test (1000
cycles). As shown in Supplementary Fig. 22, the battery retained more than 92% of its initial capacity
even after 1000 cycles, indicating its high cyclability.

We have added the result of the long-term battery cycle test as Supplementary Fig. 22 and its
explanation on page 20, lines 14–17, as below.

“As shown in Supplementary Fig. 22, a long-term battery cycle test was also performed. The battery
retained over 92% of its initial capacity even after 1000 cycles, demonstrating its high cyclability.”

**Supplementary Figure 22. The long-term battery cycle test at 10 C.**

5. I would suggest the authors investigate the structure and composition of the MOF in the electrode
and battery after being charged/discharged for several cycles (e.g. 100 cycles).

Thank you for your comment. According to your comment, to investigate the structure and
composition of the MOF of the electrode after 100 cycles of charging and discharging, we performed
PXRD and *ex situ* FT-IR measurements. Since the electrochemical behavior of UiO-66-(OH)₂ during
the charge/discharge process was identical in both half-cell and battery, we conducted PXRD and *ex*
*situ* FT-IR measurements using the anode after the battery cycle test (We also conducted PXRD
measurements using the electrode after the cycle test of the half-cell). As shown in Supplementary
Figs. 19 and 21, the structure and composition of UiO-66-(OH)₂ were well maintained even after 100
cycles.

We have added the results of PXRD and *ex situ* FT-IR measurements as Supplementary Figs. 19
and 21 and their explanation on page 20, lines 10–14, as below.

“The PXRD and *ex situ* FT-IR spectra in Supplementary Figs. 19 and 21 confirmed that the structure
and composition of UiO-66-(OH)₂ were maintained owing to its strong Zr–O bonds and the large

coordination number even after 100 cycles of the battery, supporting that both the structure and
composition of UiO-66-(OH)₂ remained unchanged.”

**Supplementary Figure 19. Confirmation that the UiO-66-(OH)₂ maintains its structure.** PXRD
patterns of UiO-66-(OH)₂/GMS/PVdF composite electrode before cycle test (black), and after 100
cycles of the half-cell (blue) and the battery (red).

**Supplementary Figure 21. Identification of compositional changes in UiO-66-(OH)₂.** *Ex situ* FT-
IR spectra of UiO-66-(OH)₂/GMS/PVdF composite electrode before the battery cycle test (black)
and after 100 cycles (red). The measurement was performed using an electrode composed of UiO-
66-(OH)₂, GMS, and PVdF (90:5:10 w/w/w), in which the amount of GMS was smaller than that
used in Figs. 2 and 3, because GMS readily absorbed infrared light. *Ex situ* FT-IR spectra were
baseline-corrected by specifying three points at 520, 570, and 700 cm⁻¹ and subtracting the linear
baseline defined by these points^{13,14}.

Thank you very much again for your positive comments. We hope that our replies are acceptable to
you.

Response to the Reviewer 3:

We appreciate very much your positive comments on our manuscript.

1-1. The manuscript's logical flow needs to be strengthened. In the Introduction, the authors mention that organic–air rechargeable batteries have an advantage over metal–air rechargeable batteries in terms of cyclability, but they also highlight the issue that organic redox-active materials tend to dissolve or degrade, which results in poor cyclability.

Thank you for your suggestion regarding the improvement of the logical flow. In order to address it, we revised the Introduction to clarify the distinction between the two different aspects of cyclability. Organic–air rechargeable batteries inherently overcame issues such as dendrite formation and carbonate clogging which were usually observed in conventional metal–air rechargeable batteries, whose cyclability typically resulted in more than 80% capacity loss after 100 cycles (reference 25 in the revised manuscript). While these issues were addressed, organic–air rechargeable batteries with organic redox-active materials as anode-active materials and acidic aqueous electrolytes still exhibited several percent capacity degradation even within 100 cycles, as shown in Supplementary Table 4, indicating that the gradual dissolution or degradation of the organic redox-active materials remained a critical issue for the cyclability. In the current work, we addressed the challenge of organic redox-active materials in acidic aqueous electrolytes by employing an acid-resistant MOF, thereby enhancing long-term cyclability and durability of organic–air rechargeable batteries. These revisions would improve the logical consistency of the Introduction and clarify the motivation and novelty of our approach.

According to your comment, we revised the explanation in the Introduction on page 4, line 17 – page 5, line 2, as below.

“These batteries inherently avoided issues such as dendrite precipitation and carbonate clogging that were commonly observed in metal–air rechargeable batteries²⁷⁻³². However, despite these advantages, organic redox-active materials often suffer from gradual dissolution or degradation in acidic aqueous electrolytes during repeated cycling, which still limits their cyclability³³.”

1-2. Furthermore, the manuscript does not clearly explain why UiO-66-(OH)₂ was chosen as the specific type of RAMOF. The rationale for selecting UiO-66-(OH)₂ as the anode-active material should be clarified in the Introduction.

Thank you for your comment. According to your comment, to clearly explain why UiO-66-(OH)₂ was chosen as the specific type of RAMOF, we have added explanations to the Introduction on page 5, lines 5–13 and on page 5, lines 14–16, as below.

“In the current work, we focus on UiO-66, which has been reported to be a crystalline porous material, a water-resistant MOF, especially an acid-resistant MOF, owing to its strong Zr–O bonds^{14,15,34} and the large coordination number³⁵. By introducing redox-active *p*-hydroquinone units (redox potential: approximately +0.1 V vs. Ag/AgCl^{27,36}), in place of benzene in the organic linker of the acid-resistant MOF, we prepare the acid-resistant RAMOF UiO-66-(OH)₂, which achieves reversible charge storage with an ideal capacity close to the theoretical capacity based on the molecular weight even in acidic aqueous electrolytes, owing to its optimized particle size, high porosity, and proton conductivity.”

“Then, by using the RAMOF as an anode-active material, an aqueous MOF–air rechargeable battery is fabricated.”

2. The authors attribute the stable structure and reversible charge/discharge behavior of UiO-66-(OH)₂ to strong Zr–O bonds and a high coordination number. However, there is a lack of direct experimental

evidence to support this claim. Spectroscopic data before and after cycling of battery should be
provided to substantiate this statement.

Thank you for your comment. According to your comment, as shown in Supplementary Fig. 21,
we provided the *ex situ* FT-IR spectra of the anode before and after the battery cycle test. The
similarity between the two spectra in points of peak position and intensity supports the stable structure
and reversible charge/discharge behavior of UiO-66-(OH)₂, which is attributed to the strong Zr–O
bonds and a high coordination number.

Therefore, we have added the measurement result as Supplementary Fig. 21 and its explanation on
page 20, lines 10–14, as below.

“The PXRD and *ex situ* FT-IR spectra in Supplementary Figs. 19 and 21 confirmed that the structure
and composition of UiO-66-(OH)₂ were maintained owing to its strong Zr–O bonds and the large
coordination number even after 100 cycles of the battery, supporting that both the structure and
composition of UiO-66-(OH)₂ remained unchanged.”

**Supplementary Figure 21. Identification of compositional changes in UiO-66-(OH)₂.** *Ex situ* FT-
IR spectra of UiO-66-(OH)₂/GMS/PVdF composite electrode before the battery cycle test (black)
and after 100 cycles (red). The measurement was performed using an electrode composed of UiO-
66-(OH)₂, GMS, and PVdF (90:5:10 w/w/w), in which the amount of GMS was smaller than that
used in Figs. 2 and 3, because GMS readily absorbed infrared light. *Ex situ* FT-IR spectra were
baseline-corrected by specifying three points at 520, 570, and 700 cm⁻¹ and subtracting the linear
baseline defined by these points^{13,14}.

3. I recommend that the authors include some *in-situ* characterization during the battery cycling
process to further strengthen the manuscript and enhance its credibility.

Thank you for your comment. In order to further strengthen the manuscript and enhance its
credibility, we performed *in situ* FT-IR analysis in addition to *ex situ* FT-IR analysis. Since the
electrochemical behavior of UiO-66-(OH)₂ during the charge/discharge process was identical in both
half-cell and battery, we conducted *ex situ* and *in situ* FT-IR measurements using the electrode in the
half-cell. By focusing on the initial state of the structure of UiO-66-(OH)₂, as shown in
Supplementary Fig. 11 (DFT calculations), we found that the optimized structure of the cluster of
UiO-66-(OH)₂ had hydrogen bonds between hydroxyl and carboxy groups. Upon oxidation at around
+0.7 V vs. Ag/AgCl (Supplementary Fig. 9), the formation of the C=O was confirmed by *ex situ* FT-
IR analysis. Based on *in situ* FT-IR analysis (Supplementary Fig. 10) and previous literature (redox
reactions of non-hydrogen-bonded *p*-hydroquinone (references 47 and 48 in the revised manuscript)),
the oxidation peaks at around +0.3 V and +0.7 V vs. Ag/AgCl were attributed to the oxidation of

non-hydrogen-bonded and hydrogen-bonded C–(OH) of *p*-hydroquinone, respectively. Furthermore,
we conducted DFT calculations to identify redox sites of the oxidation state of UiO-66-(OH)₂,
following the methodologies reported in previous works (references 56 and 57 in the revised
manuscript). The molecular electrostatic potential (MESP) mapping revealed that oxygen atoms of
the *p*-benzoquinone moiety exhibited a significant negative MESP value, indicating their suitability
for proton storage. Based on *ex situ* and *in situ* FT-IR analyses and DFT calculations, we confirmed
the redox reaction between *p*-hydroquinone and *p*-benzoquinone in the UiO-66-(OH)₂.

We have added the measurement result as Supplementary Fig. 10 and its explanation on page 12,
line 1 – page 14, line 6, as below.

“In order to investigate the details of the charge storage mechanism of UiO-66-(OH)₂, as shown in
Supplementary Figs. 9 and 10, we performed *ex situ* and *in situ* FT-IR analyses. By focusing on the
initial state of the structure of UiO-66-(OH)₂, as shown in Supplementary Fig. 11 and Supplementary
Table 5, we found that the density functional theory (DFT)-optimized structure of the cluster of UiO-
66-(OH)₂ exhibited an O···O distance of 2.48 Å⁴⁹, suggesting that the initial state of the structure of
UiO-66-(OH)₂ had hydrogen bonds between the protons of C–(OH) of *p*-hydroquinone and carboxy
groups. As shown in Fig. 2a and Supplementary Fig. 9a, an irreversible oxidation peak appeared at
around +0.7 V vs. Ag/AgCl upon sweeping the potential in the positive direction from +0.50 V to
+0.90 V vs. Ag/AgCl. As shown in Supplementary Figs. 9a and 9b, in the *ex situ* FT-IR spectrum
after applying the potential at +0.90 V vs. Ag/AgCl, a new peak appeared at 1638 cm⁻¹, derived from
C=O⁵⁰, which indicated that the new peak was attributed to the formation of C=O by the oxidation
(Supplementary Fig. 9c). This attribution was further supported by the finding that the oxidation
potential shift to the positive direction was presumably caused by hydrogen-bond formation, as
reported in previous works^{51,52}. In addition, as shown in Supplementary Fig. 10b, the peak intensity
of 1235 cm⁻¹ at the initial state of the red solid-line spectrum, attributed to C–(OH) of *p*-
hydroquinone⁵³⁻⁵⁵, decreased upon oxidation, resulting in the purple solid-line spectrum, thereby
suggesting the oxidation of hydrogen bonded C–(OH) of *p*-hydroquinone. From the above results,
as shown in Supplementary Fig. 10c, the oxidation peak at around +0.7 V vs. Ag/AgCl was attributed
to the oxidation of hydrogen-bonded C–(OH) of *p*-hydroquinone to C=O of *p*-benzoquinone. After
that, as shown in Supplementary Fig. 10a, upon sweeping the potential from +0.90 V to –0.20 V vs.
Ag/AgCl, we observed a reduction peak at around 0.0 V vs. Ag/AgCl and an increase of the peak
intensity from the purple solid- to the blue solid-line spectra (Supplementary Fig. 10b). As shown in
Supplementary Fig. 12 and Supplementary Table 6, the molecular electrostatic potential (MESP)
map suggested that C=O of *p*-benzoquinone in the oxidation state of UiO-66-(OH)₂ would be the
reduction site for proton storage owing to the strongly negative MESP value of the oxygen atoms in
*p*-benzoquinone^{56,57}. These results indicated that the reduction peak around 0.0 V vs. Ag/AgCl
(Supplementary Fig. 10b) was attributed to the reduction of C=O of *p*-benzoquinone to C–(OH) of
*p*-hydroquinone (Supplementary Fig. 10c). After that, upon sweeping the potential from –0.20 V to
+0.90 V vs. Ag/AgCl (Supplementary Fig. 10a), as shown in Supplementary Fig. 10b, we observed
two kinds of oxidation peaks at around +0.3 V (polarized) and +0.7 V vs. Ag/AgCl, and observed a
decrease in peak intensity from the blue solid- via red dotted- to purple dotted-line spectra. Based on
these peak intensity changes (Supplementary Figs. 10b and 10c) and previous literature (redox
reactions of non-hydrogen-bonded *p*-hydroquinone^{47,48}), the oxidation peaks around +0.3 V and +0.7
V vs. Ag/AgCl were attributed to non-hydrogen-bonded and hydrogen-bonded C–(OH) of *p*-
hydroquinone, respectively. From the above analyses, as shown in Fig. 2a, the oxidation peak at
around +0.7 V vs. Ag/AgCl was attributed to the oxidation of hydrogen-bonded C–(OH) of *p*-
hydroquinone (Supplementary Fig. 10c), and the redox peak in the range of –0.1–+0.4 V vs.
Ag/AgCl was attributed to the redox reaction of non-hydrogen-bonded C–(OH) of *p*-hydroquinone

(Supplementary Fig. 10c).”

**Supplementary Figure 9. Identification of conformationally changed sites in UiO-66-(OH)₂.** (a,
b) *Ex situ* FT-IR spectra of UiO-66-(OH)₂ (black) and UiO-66-(OH)₂ after applying a potential of
+0.90 V vs. Ag/AgCl for 2 h in a 0.05 M H₂SO₄ aqueous solution. This spectrum was obtained by
converting the transmittance data using the following equation. “ $A = \log(1/T)$ (A: Absorbance, T:
Transmittance)” Potentiostatic electrolysis was performed using a UiO-66-(OH)₂ electrode, which
was fabricated by drop-casting a slurry of UiO-66-(OH)₂ and *N*-methyl-2-pyrrolidone onto a glassy
carbon electrode and drying at 120°C for 2 h. After performing potentiostatic electrolysis, the
electrode was immersed in water to remove the electrolyte for measuring the *ex situ* FT-IR spectrum.
The spectra were normalized by the peaks at 1590 cm⁻¹, attributed to the COO⁻ of the organic
linkers¹⁶, which remained unchanged before and after the reaction. (c) Molecular structural changes
estimated from the difference of *ex situ* FT-IR spectra.

Supplementary Figure 10. Identification of conformationally changed sites in UiO-66-(OH)₂

based on *in situ* measurement.

(a) Different charge/discharge states selected from the cyclic voltammogram and (b) corresponding *in situ* FT-IR spectra. The measurement was performed using an electrode composed of UiO-66-(OH)₂ and SWNT (5:1 w/w), in which the carbon ratio is smaller than that described in the Methods. Therefore, a part of the C-(OH) groups of *p*-hydroquinone contributed to the redox reaction. Since a strong peak of H₂O was observed at around 1600 cm⁻¹, we focused on the peak at around 1235 cm⁻¹, corresponding to the C-(OH) group of *p*-hydroquinone¹⁵⁻¹⁷. *In situ* FT-IR spectra were baseline-corrected by specifying two points at 1210 and 1260 cm⁻¹ and subtracting the linear baseline defined by these points¹³⁻¹⁴. (c) The estimated structural changes during charge/discharge measurements. Once the electrode was oxidized, the redox peak in the range of -0.1→+0.4 V vs. Ag/AgCl became predominant compared to the irreversible oxidation peak at +0.7 V vs. Ag/AgCl (Fig. 2a), presumably because it took time for the hydrogen bonds to form¹⁸.

4-1. In Fig. 2(a), the CV curve of the first cycle shows significant differences compared to the subsequent cycles. Please provide an explanation for this observation.

Thank you for your comment. In order to further explain the significant differences between the first cycle to the subsequent cycles, as shown in Supplementary Figs. 9 and 10, we performed *in situ* FT-IR analysis in addition to *ex situ* FT-IR analysis. As shown in Supplementary Fig. 11, in the initial state, hydrogen bonds between hydroxyl and carboxy groups were formed, which was corroborated by DFT calculations. Upon oxidation at around +0.7 V vs. Ag/AgCl (Supplementary Fig. 9), the formation of the C=O was confirmed by *ex situ* FT-IR analysis. Furthermore, once the electrode was oxidized, the redox peak in the range of -0.1→+0.4 V vs. Ag/AgCl became predominant compared to the irreversible oxidation peak at around +0.7 V vs. Ag/AgCl. Based on *in situ* FT-IR analysis (Supplementary Fig. 10) and previous literature (redox reactions of non-hydrogen-bonded *p*-hydroquinone (references 47 and 48 in the revised manuscript)), the oxidation peaks at around +0.3 V and +0.7 V vs. Ag/AgCl were attributed to the oxidation of non-hydrogen-bonded and hydrogen-bonded C-(OH) of *p*-hydroquinone, respectively. From the above, the irreversible oxidation peak in the first cycle of Fig. 2a was the oxidation of hydrogen-bonded C-(OH) of *p*-hydroquinone, and, in the subsequent cycles, the reversible redox peak in the range of -0.1→+0.4 V vs. Ag/AgCl appeared,

which derived from the redox reaction of non-hydrogen-bonded C–(OH) of *p*-hydroquinone.

According to your comment, we have added more detailed explanations of the CV curve observed
in the first cycle and subsequent cycles, as below.

On page 11, lines 12–14. “In the first cycle, the potential was swept from +0.50 V vs. Ag/AgCl to
5 –0.20 V vs. Ag/AgCl and then swept back to +0.50 V vs. Ag/AgCl.”

On page 11, line 17 – page 12, line 1. “However, as shown in Fig. 2a, upon sweeping the potential
in the positive direction from +0.50 V vs. Ag/AgCl to +0.90 V vs. Ag/AgCl, an irreversible oxidation
peak appeared at around +0.7 V vs. Ag/AgCl, and a redox peak emerged in the range of –0.1–+0.4
V vs. Ag/AgCl.”

On page 14, lines 2–6. “From the above analyses, as shown in Fig. 2a, the oxidation peak at around
+0.7 V vs. Ag/AgCl was attributed to the oxidation of hydrogen-bonded C–(OH) of *p*-hydroquinone
(Supplementary Fig. 10c), and the redox peak in the range of –0.1–+0.4 V vs. Ag/AgCl was
attributed to the redox reaction of non-hydrogen-bonded C–(OH) of *p*-hydroquinone (Supplementary
Fig. 10c).”

**Supplementary Figure 9. Identification of conformationally changed sites in UiO-66-(OH)₂.** (a,
b) *Ex situ* FT-IR spectra of UiO-66-(OH)₂ (black) and UiO-66-(OH)₂ after applying a potential of
+0.90 V vs. Ag/AgCl for 2 h in a 0.05 M H₂SO₄ aqueous solution. This spectrum was obtained by
converting the transmittance data using the following equation. “A = log (1/T) (A: Absorbance, T:
Transmittance)” Potentiostatic electrolysis was performed using a UiO-66-(OH)₂ electrode, which
was fabricated by drop-casting a slurry of UiO-66-(OH)₂ and *N*-methyl-2-pyrrolidone onto a glassy

carbon electrode and drying at 120°C for 2 h. After performing potentiostatic electrolysis, the
 electrode was immersed in water to remove the electrolyte for measuring the *ex situ* FT-IR spectrum.
 The spectra were normalized by the peaks at 1590 cm⁻¹, attributed to the COO⁻ of the organic
 linkers¹⁶, which remained unchanged before and after the reaction. (c) Molecular structural changes
 estimated from the difference of *ex situ* FT-IR spectra.

 **Supplementary Figure 10. Identification of conformationally changed sites in UiO-66-(OH)₂**
 **based on *in situ* measurement.** (a) Different charge/discharge states selected from the cyclic
 voltammogram and (b) corresponding *in situ* FT-IR spectra. The measurement was performed using
 an electrode composed of UiO-66-(OH)₂ and SWNT (5:1 w/w), in which the carbon ratio is smaller
 than that described in the Methods. Therefore, a part of the C-(OH) groups of *p*-hydroquinone
 contributed to the redox reaction. Since a strong peak of H₂O was observed at around 1600 cm⁻¹, we
 focused on the peak at around 1235 cm⁻¹, corresponding to the C-(OH) group of *p*-hydroquinone¹⁵⁻
 ¹⁷. *In situ* FT-IR spectra were baseline-corrected by specifying two points at 1210 and 1260 cm⁻¹ and
 subtracting the linear baseline defined by these points¹³⁻¹⁴. (c) The estimated structural changes during
 charge/discharge measurements. Once the electrode was oxidized, the redox peak in the range of
 18 -0.1→+0.4 V vs. Ag/AgCl became predominant compared to the irreversible oxidation peak at +0.7
 V vs. Ag/AgCl (Fig. 2a), presumably because it took time for the hydrogen bonds to form¹⁸.

**Supplementary Figure 11. The optimized structure of the cluster of UiO-66-(OH)₂.** The O...O
distance was calculated to be 2.48 Å, which indicated the formation of hydrogen bonds between
hydroxyl and carboxy groups¹⁹.

4-2. Additionally, from the 1st to the 26th cycle, a polarization is observed in the CV curves. Please
elaborate on the possible causes of this polarization.

Thank you for your comment. In the previous work (reference 46 in the revised manuscript), density
functional theory (DFT) calculations suggested that the semiquinone state was stable at pH < 1 with
respect to disproportionation. This stability led to two one-electron redox steps instead of one two-
electron process. As a result, as shown in Fig. 2a, polarization was observed in the oxidation process.

We have added the explanation of the polarization on page 10, line 5 – page 11, line 1, as below.

“As shown in Fig. 2a, a polarization was observed. The previous work reported that the semiquinone
state is stabilized at pH < 1, according to DFT calculations, leading to two one-electron oxidation
steps rather than a single two-electron step⁴⁶. In addition, since the conversion of quinone in its neutral
state to quinone radical anion was unfavorable, the reduction proceeded via the protonated
intermediate⁴⁶. These factors contributed to the polarization observed in the oxidation process.”

5. The authors tested the EIS of UiO-66-(OH)₂ and reported an ion conductivity of 2.18×10⁻⁶ S/cm.
Besides proton conductivity, do other types of ions contribute to this ion conductivity?

Thank you for your comment. Since UiO-66-(OH)₂ had no residual metal ions, under the humidified
conditions used in the EIS measurement, the possible charge carriers were limited to protons (H⁺)
and hydroxide ions (OH⁻). The mobility of hydroxide ions (OH⁻) when they move in nanopores via
the vehicle mechanism has been reported to be approximately two orders of magnitude lower than
that of protons (H⁺), owing to the large activation energy of partial dehydration of hydroxide ions
(OH⁻) that disrupts their stable hydration shell (reference 20 in the revised supplementary
information). Therefore, the contribution of hydroxide ions (OH⁻) to the ionic conduction of UiO-
66-(OH)₂ was considered to be much smaller than that of protons (H⁺), and the measured ionic
conductivity of 2.18×10⁻⁶ S cm⁻¹ was concluded to be primarily due to proton conduction, not to
other types of ions.

We have added the explanation of the ion conductivity to the caption of Supplementary Fig. 15, as
below.

“Since UiO-66-(OH)₂ had no residual metal ions, under humidified conditions, the possible ionic

species are limited to protons (H^+) and hydroxide ions (OH^-). The mobility of hydroxide ions (OH^-)
in nanopores via the vehicle mechanism has been reported to be approximately two orders of
magnitude lower than that of protons (H^+), due to partial dehydration of hydroxide ions (OH^-) that
disrupts their stable hydration shell and requires a large activation energy^{20,21}. Therefore, the
contribution of hydroxide ions (OH^-) to the ionic conduction of $UiO-66-(OH)_2$ was considered to be
much smaller than that of protons (H^+), and the measured ionic conductivity of $2.18 \times 10^{-6} \text{ S cm}^{-1}$
was primarily due to proton conduction, not to other types of ions²².”

6. Additionally, after the high-rate charging/discharging (e.g., 45 C), does the structure of $UiO-66-$
$(OH)_2$ undergo any changes? Can the authors provide relevant characterization to confirm its stability,
as this would be crucial for demonstrating its high durability and stability.

Thank you for your comment. According to your comment, we performed PXRD measurements
of the electrode after 100 cycles at a high rate of 42 C in the half-cell. As shown in Supplementary
Fig. 19, the PXRD pattern confirmed that the structure of $UiO-66-(OH)_2$ was maintained even after
100 cycles, demonstrating its durability and stability.

We have added the result of PXRD as Supplementary Fig. 19 and its explanation on page 16, line
18 – page 17, line 2, as below.

“The PXRD in Supplementary Fig. 19 confirmed that the structure of $UiO-66-(OH)_2$ was
maintained owing to its strong Zr–O bonds and the large coordination number even after 100 cycles
in the half-cell, showing its high structural stability.”

**Supplementary Figure 19. Confirmation that the $UiO-66-(OH)_2$ maintains its structure.** PXRD
patterns of $UiO-66-(OH)_2$ /GMS/PVdF composite electrode before cycle test (black), and after 100
cycles of the half-cell (blue) and the battery (red).

7. The cycling data of battery presented in the manuscript is impressive, with minimal degradation
observed at current rates ranging from 5 C to 45 C. However, how much further can the current rate
be increased before noticeable degradation of the battery occurs?

Thank you for your comment. According to your comment, we performed high-rate
charge/discharge measurements of the battery at 60 C. As shown in Supplementary Fig. 23, the
battery exhibited a discharge capacity of 102.8 mAh g⁻¹, corresponding to 60 % of the theoretical
capacity, and noticeable degradation of the battery performance was observed at 60 C.

We have added the result as Supplementary Fig. 23 and its explanation on page 20, line 17 – page
21, line 1, as below.

“In addition, as shown in Fig. 3d, the battery exhibited high-rate capabilities, achieving a discharge
capacity of 157.3 mAh g⁻¹ (92% of the theoretical capacity) even at 45 C, and, as shown in
Supplementary Fig. 23, it retained a discharge capacity of 102.8 mAh g⁻¹ (60% of the theoretical
capacity) at 60 C.”

**Supplementary Figure 23. High-rate capability of the battery.** Charging (black)/discharging (red)
curves of the battery at 60 C. The dotted line represents the theoretical capacity based on the
molecular weight of UiO-66-(OH)₂ (171.9 mAh g⁻¹).

Thank you very much again for your productive comments. We hope that our replies are acceptable
to you.

**Response to the Reviewer 4:**

We appreciate very much your positive comments on our manuscript.

1. In the Introduction section on page 3, the sentence "However, MOFs are usually structurally weak
in aqueous solutions, especially in acidic aqueous solutions, owing to their coordination bonds,
making their application in charge-storage devices challenging." requires further elaboration. It would
be beneficial to provide a more detailed explanation of why MOF's coordination bond is vulnerable
in the acidic aqueous solution.

Thank you for your comment. According to your comment, we have added a more detailed
explanation of why MOFs' coordination bonds are vulnerable in acidic aqueous solutions on page 3,
lines 7–11, as below.

“However, since acidic protons can cleave coordination bonds in MOFs by promoting hydrolysis
of the metal-organic linker bonds¹³, MOFs are usually structurally weak in aqueous solutions,
especially in acidic aqueous solutions, making their application in aqueous charge-storage devices
challenging^{14,15}.”

2. If possible, it would be beneficial to include long-term cycling data beyond 100 cycles. The data
presented in Supplementary Table 3 demonstrates that this work exhibits superior performance
compared to other samples up to 100 cycles. However, based on the current data, it is unclear whether
this trend persists over an extended cycling period.

Thank you for your comment. According to your comment, we conducted a long-term cycling test
(1000 cycles) on the half-cell. As shown in Supplementary Fig. 20, the half-cell retained more than
95% of its initial capacity even after 1000 cycles, demonstrating the high durability of UiO-66-(OH)₂
as the anode-active material.

We have added the result of the long-term cycle test of the electrode as Supplementary Fig. 20 and
its explanation on page 17, lines 2–5, as below.

“As shown in Supplementary Fig. 20, a long-term cycle test of the electrode was also performed.
The UiO-66-(OH)₂/GMS/PVdF composite electrode retained over 95% of its initial capacity even
after 1000 cycles, demonstrating its high cyclability.”

**Supplementary Figure 20. Long-term cycle test of the electrode at 45 C.**

3. On page 15, in Fig. 3 Schematic and performance of the aqueous MOF–air rechargeable battery.
Schematic diagrams of the (a) charging/(b) discharging of the aqueous MOF–air rechargeable battery,
it appears that current collector is missing from the figure. If this omission was intentional, it would
be helpful to provide an explanation.

Thank you very much for pointing out that the current collectors were missing. According to your

comment, we have added the current collector to the schematic diagrams of the (a) charging/(b) discharging of the aqueous MOF–air rechargeable battery in Figs. 3a and 3b on page 18.

Fig. 3 Schematic and performance of the aqueous MOF–air rechargeable battery. Schematic diagrams of the (a) charging/(b) discharging of the aqueous MOF–air rechargeable battery.

4. In the Supplementary Figure 13, MOF active material appears to be easily peeled off from glassy carbon electrode. It would be important to provide an explanation of whether this could pose an issue during battery operation.

Thank you for your comment. According to your comment, we have added a detailed explanation of the electrode to the caption of Supplementary Fig. 24, as below.

“The UiO-66-(OH)₂/GMS/PVdF composite electrode was peeled off with a spatula. The electrode remained firmly adhered during actual battery operation, and no detachment was observed during electrochemical testing.”

5. On page 19 of the main text, the sentence mentioning " As described in Experimental Section 2.2, although the yield of UiO-66-(OH)₂-R was still low, the recycling yield could be improved by investigating decomposition and reconstruction conditions (e.g. solvent and modulator) in our continuous work." is present. While the detailed procedure is described in Experimental Section 2.2, it would be helpful to explicitly state the yield as a numerical value in the main text as well.

Thank you for your comment. According to your comment, we explicitly stated the yield as a numerical value in the main text on page 23, lines 5–8, as below.

“As described in the Experimental Section 2.2, although the yield of UiO-66-(OH)₂-R (approximately 10%) was still low, the recycling yield could be improved by investigating decomposition and reconstruction conditions (e.g., solvent and modulator) in our continuous work.”

Thank you very much again for your positive comments. We hope that our replies are acceptable to you.

Response to the Reviewer 1:

We appreciate very much your positive comments on our manuscript.

0. The authors' response is generally sufficient, reasonable, and scientifically based. By supplementing the paper with experimental data (such as concentration gradient SEM, FT-IR comparisons, in situ/ex situ FT-IR, and DFT calculations) and theoretical analysis, they have enhanced the paper's completeness and persuasiveness, further strengthening its innovativeness and scientific nature.

Thank you very much for the positive comment.

1. The use of "-" ("-") in various forms throughout the paper makes some data less visually appealing. For example, on Page 12, "-0.1+0.4 V vs. Ag/AgCl" and Page 12, "-0.1+0.4 V vs. Ag/AgCl," it is recommended that spaces be added or replaced with "~" to separate word.

Thank you very much for pointing out the visual issue regarding the use of “-”. According to your comment, we have revised all range notations by adding spaces before and after “-” to improve readability throughout the revised manuscript and supplementary information.

On page 4, lines 11–15, in the revised manuscript. “In addition, highly concentrated basic aqueous electrolytes (6 – 7 M KOH aqueous solution) are often used for efficient ionic conduction, which causes carbonate clogging owing to reactions of the electrolyte with atmospheric CO₂, leading to lower cyclability of the batteries²⁶.”

On page 9, lines 7–10, in the revised manuscript. “However, as shown in Fig. 2a, upon sweeping the potential in the positive direction from +0.50 V vs. Ag/AgCl to +0.90 V vs. Ag/AgCl, an irreversible oxidation peak appeared at around +0.7 V vs. Ag/AgCl, and a redox peak emerged in the range of –0.1 – +0.4 V vs. Ag/AgCl.”

On page 11, lines 11–15, in the revised manuscript. “From the above analyses, as shown in Fig. 2a, the oxidation peak at around +0.7 V vs. Ag/AgCl was attributed to the oxidation of hydrogen-bonded C–(OH) of *p*-hydroquinone (Supplementary Fig. 10c), and the redox peak in the range of –0.1 – +0.4 V vs. Ag/AgCl was attributed to the redox reaction of non-hydrogen-bonded C–(OH) of *p*-hydroquinone (Supplementary Fig. 10c).”

On page 17, lines 5–8, in the revised manuscript. “Figs. 4a – 4d and Supplementary Tables 3 and 4 summarize the advantages of the aqueous MOF–air rechargeable battery compared to aqueous MOF-based rechargeable batteries¹⁷⁻²² and aqueous organic–air rechargeable batteries^{27-31,38,56,62,68-70}.”

On page 21, lines 5–6, in the revised manuscript. “The mass loading of UiO-66-(OH)₂ was adjusted to approximately 0.1 – 1.0 mg.”

On page 3, line 20 – page 4, line 2, in the revised supplementary information. “Alternating current (AC) impedance measurements were performed using the ALS 760E dual electrochemical analyzer (BAS Ltd.) in the frequency range 10⁰ – 10⁶ Hz with 0.005 V (amplitude voltage).”

On page 6, lines 8–10, in the revised supplementary information. “The combustion reaction (280 – 450°C) of defect-free UiO-66-(OH)₂ is represented by the following equation⁶: $Zr_6O_6(C_8H_4O_6)_6 + 39O_2 \rightarrow 6ZrO_2 + 48CO_2 + 12H_2O$ ”

On page 7, lines 1–2, in the revised supplementary information. “The mass loading of UiO-66-(OH)₂ was adjusted to approximately 0.1 – 1.0 mg.”

On page 7, lines 6–7, in the revised supplementary information. “The mass loading of UiO-66-(OH)₂ was adjusted to approximately 0.1 – 1.0 mg.”

On page 19, line 12 – page 20, line 1, in the revised supplementary information. “Once the electrode was oxidized, the redox peak in the range of –0.1 – +0.4 V vs. Ag/AgCl became predominant compared to the irreversible oxidation peak at +0.7 V vs. Ag/AgCl (Fig. 2a), presumably because it took time for the hydrogen bonds to form¹⁸.”

2. The order of the supporting figures is confusing. On Page 6, the title of Figure 1 refers to "Supplementary Fig. 5," but Supplementary Fig. 1 is not mentioned until Page 6. Please ensure that all supporting information is included.

Thank you very much for your comment. We have revised the order and position of the supplementary figures in the revised manuscript in accordance with the editor's editorial requests. In addition, we have carefully checked the revised supplementary information to ensure that all supporting information (figures and tables) is properly included.

Response to the Reviewer 2:

The authors have addressed the reviewers' comments properly; therefore, I suggest acceptance of the manuscript.

Thank you very much for your positive comment.

Response to the Reviewer 3:

The authors have well addressed the questions, and therefore I recommend it for publication without change.

Thank you very much for your positive comment.

Response to the Reviewer 4:

The revisions have been thoroughly reviewed and appropriately reflect our suggestions. We are satisfied with the changes and have no further comments.

Thank you very much for your positive comment.

Recommendation: Publish after revisions noted.

Comments:

This paper is highly significant as it presents the first case of using graphene mesosponge (GMS) as a conductive additive for redox-active MOFs (RAMOFs) and also marks the first instance of utilizing RAMOF as an anode-active material in aqueous rechargeable batteries.

This paper was well organized, and the results are interesting, thus I recommend publishing the manuscript in Nature Communications after revision.

1. In the Introduction section on page 3, the sentence "However, MOFs are usually structurally weak in aqueous solutions, especially in acidic aqueous solutions, owing to their coordination bonds, making their application in charge-storage devices challenging." requires further elaboration. It would be beneficial to provide a more detailed explanation of why MOF's coordination bond is vulnerable in the acidic aqueous solution.

2. If possible, it would be beneficial to include long-term cycling data beyond 100 cycles. The data presented in Supplementary Table 3 demonstrates that this work exhibits superior performance compared to other samples up to 100 cycles. However, based on the current data, it is unclear whether this trend persists over an extended cycling period.

3. On page 15, in Fig. 3 Schematic and performance of the aqueous MOF–air rechargeable battery. Schematic diagrams of the (a) charging/(b) discharging of the aqueous MOF–air rechargeable battery, it appears that current collector is missing from the figure. If this omission was intentional, it would be helpful to provide an explanation.

4. In the Supplementary Figure 13, MOF active material appears to be easily peeled off from glassy carbon electrode. It would be important to provide an explanation of whether this could pose an issue during battery operation.

5. On page 19 of the main text, the sentence mentioning " As described in Experimental

Section 2.2, although the yield of UiO-66-(OH)₂-R was still low, the recycling yield could be improved by investigating decomposition and reconstruction conditions (e.g. solvent and modulator) in our continuous work." is present. While the detailed procedure is described in Experimental Section 2.2, it would be helpful to explicitly state the yield as a numerical value in the main text as well.